# Harnessing the Potential of Walnut Leaves from Nerpio: Unveiling Extraction Techniques and Bioactivity Through *Caenorhabditis elegans* Studies

**DOI:** 10.3390/foods14061048

**Published:** 2025-03-19

**Authors:** Amel Hamdi, Miguel Angel Córdoba-Rojano, Jose Manuel Monje-Moreno, Elisa Guillén-Izquierdo, Rocío Rodríguez-Arcos, Ana Jiménez-Araujo, Manuel Jesús Muñoz-Ruiz, Rafael Guillén-Bejarano

**Affiliations:** 1Instituto de la Grasa, Consejo Superior de Investigaciones Científicas (CSIC), Pablo de Olavide University Campus, Building 46, Carretera de Utrera Km 1, 41013 Seville, Spain; miguelcordobarojano@gmail.com (M.A.C.-R.); eligui99@hotmail.com (E.G.-I.); rrodri@ig.csic.es (R.R.-A.); araujo@ig.csic.es (A.J.-A.); 2Molecular Biology and Biochemical Engineering Department, Centro Andaluz de Biología del Desarrollo (CABD), University Pablo de Olavide (UPO), CSIC/UPO/JA, Carretera de Utrera Km 1, 41013 Seville, Spain; jmmonmor@upo.es (J.M.M.-M.); mmunrui@upo.es (M.J.M.-R.)

**Keywords:** walnut leaves, Gran Jefe variety, Nerpio, phenolic compounds, extraction methods, water, ethanol 80%, bioactivities, free phenolics, *C. elegans*

## Abstract

This study used *Juglans regia* leaves from the Gran Jefe variety; this indigenous cultivar from Nerpio is highly valued for its quality and distinct characteristics. This type of walnut is traditionally cultivated in the region and is noted for its organoleptic properties and adaptation to local climatic conditions. Two solvents were tested to determine the optimal extraction conditions for phenolic compounds: 80% ethanol and water. Direct homogenization with an Ultra-Turrax, direct ultrasound, and indirect ultrasound treatments were compared for ethanol extraction. Water extractions were conducted using direct and indirect ultrasound, infusion, and decoction. Compared to water extraction, 80% ethanol proved to be more efficient. Extracting phenolic compounds from ’Gran Jefe’ walnut leaves was most effective when using direct extraction methods without either ultrasound assistance or indirect ultrasound treatment. The main compounds identified were *trans*-3-caffeoylquinic acid and quercetin-3-hexoside isomer 1. The ethanolic extract obtained through direct extraction was selected to study further the bioactivities of ’Gran Jefe’ walnut leaves using *C. elegans* as an in vivo model. Results indicated that the leaf extract enhanced thermal and oxidative stress resistance, promoted fertility, and exhibited neuroprotective effects in models of Alzheimer’s and Parkinson’s diseases. The observed bioactivities were attributed to the free phenolics present in the ethanolic extract.

## 1. Introduction

The Persian walnut, *J. regia*, is a multifunctional and remarkable walnut tree that has been valued for centuries for its edible nuts [1]. However, its leaves are often overlooked and underestimated in terms of their potential applications in various industries. The presence of bioactive compounds in these leaves, including phenolic acids [2,3], flavonoids [3,4,5], naphthoquinones [3,6] tannins [7,8] and dietary fiber [9], contributes to their numerous beneficial properties [5,10,11,12]. In the realm of food, *J. regia* leaves can be utilized in different culinary preparations. Traditionally, they have been utilized as natural flavor enhancers in culinary applications, providing a subtle and nutty taste to certain dishes. Additionally, these leaves can be brewed into an herbal tea, offering a delightful flavor and potential health benefits [2,13].

Medicinally, *J. regia* leaves have been recognized for their therapeutic properties. They possess antioxidant [4,10,13,14,15,16,17,18] and anti-inflammatory [19,20] properties, making them a potential candidate for natural remedies and herbal medicine. Extracts derived from these leaves have shown promising results in various studies, demonstrating their potential to alleviate symptoms related to oxidative stress, inflammation, and certain chronic diseases [11,19]. Moreover, the leaves of *J. regia* are known for their antimicrobial and antifungal activities [5,10,16,20,21]. The presence of natural compounds like juglone and quinone derivatives in the leaves contributes to their antibacterial and antifungal properties [22,23]. These properties make *J. regia* leaves an effective natural remedy for various infections, including dermatophytes and pathogens affecting the urinary and respiratory systems.

In traditional medicine practices, *J. regia* leaves have also been utilized to aid digestion and improve gastrointestinal health. The leaves possess astringent properties that can help soothe digestive issues such as diarrhea and dysentery [9,24]. Furthermore, they have been used to promote liver health and support detoxification processes within the body [25].

Studies have shown promising results in utilizing *J. regia* leaves as a natural remedy for managing diabetes. The phenolic compounds and flavonoids present in these leaves have been found to have a positive impact on blood glucose levels [11,26]. These compounds work synergistically to enhance insulin sensitivity, stimulate glucose uptake by cells, and inhibit the action of enzymes responsible for carbohydrate digestion [27,28,29].

Moreover, several works have shown encouraging findings when exploring the therapeutic properties of *J. regia* leaves in Alzheimer’s and Parkinson’s disease treatment. Abundant in antioxidants, these leaves provide crucial protection to neural cells by combating oxidative stress and inflammation-induced damage [15]. One key mechanism by which *J. regia* leaves may combat Alzheimer’s is their ability to inhibit the aggregation and accumulation of beta-amyloid plaques in the extracellular matrix of cerebral tissue. Beta-amyloid plaques are abnormal protein aggregates considered a characteristic neuropathological feature of Alzheimer’s disease. They disrupt normal brain function and lead to the degeneration of neurons [30,31,32].

Previous studies [2,4,6,16] identified various hydroxycinnamic acids (3-caffeoylquinic, 5-caffeoylquinic, 3-*p*-coumaroylquinic, and 4-*p*-coumaroylquinic acids) and flavonoids (quercetin-3-galactoside, quercetin-3-arabinoside, quercetin-3-xyloside, quercetin-3-rhamnoside, quercetin-3-pentoside derivatives, kaempferol-3-pentoside, myricetin-3-glucoside and myricetin-3-pentoside) in ethanol and aqueous extracts of walnuts leaves. In addition, juglone, also known as 5-hydroxy-1,4-naphthoquinone, is recognized as the distinctive molecule found in trees of the *Juglans* spp. family. The presence of this compound has been documented in freshly grown leaves of walnut trees [3].

*Caenorhabditis elegans* is a microscopic roundworm that has been significant in scientific research for decades. Despite its simple appearance, this small organism has captivated the attention of scientists worldwide, earning its place as a key model organism in studies involving the potential biological activity of plant extracts. *C. elegans* shares a surprising degree of genetic similarity to humans. Approximately 40% of its genes have human orthologs, meaning they have similar counterparts in our genome. Over the years, *C. elegans* has contributed invaluable insights into numerous human diseases, including Alzheimer’s, Parkinson’s, Huntington’s, cancer, diabetes, and other aging-related diseases. Its genetic tractability allows researchers to manipulate and study specific genes, unravelling the intricate mechanisms underlying these diseases [33].

Since walnut leaves possess diverse biological effects and can potentially be targeted for producing high-quality plant-based ingredients in the food, beverage, and dietary supplement industries, it is important to produce liquid extracts using aqueous and/or ethanolic solvents. This study aims to determine the optimal extraction conditions for the native Gran Jefe variety of *J. regia* leaves from Nerpio, Spain, which is highly valued for its quality and distinct characteristics. Traditionally cultivated in the region, this variety is noted for its organoleptic properties and adaptation to local climatic conditions. The study will also investigate the differences in phenolic composition based on the extraction conditions and evaluate their effects on oxidative stress resistance and infertility issues associated with diabetes, as well as Alzheimer’s and Parkinson’s diseases, using *C. elegans* as an in vivo model.

## 2. Materials and Methods

### 2.1. Plant Material

Leaves of the walnut tree (*J. regia*), variety ‘Gran Jefe’, were collected on 30 August 2022, from a plantation in Nerpio, Spain. The fresh leaf samples were vacuum-sealed to prevent oxidation and transported to the laboratory. Upon arrival, the samples were maintained at −20 °C prior to analytical procedures.

### 2.2. Extraction Procedure

Two solvents, 80% ethanol (E) and water (W), were used. In both cases, the samples were cut into small pieces and homogenized with an Ultra-Turrax T25 (Janke & Kukel/IKA Labortechnik, Munich, Germany) at a 10:1 (*v*/*w*) solvent-to-solid ratio. The homogenates were subsequently separated from the solid residue via Whatman No. 1 filter paper (Anoia, Barcelona, España) and maintained at −20 °C before analytical procedures (samples E and W).

Two techniques were employed to investigate the impact of ultrasound treatment: indirect ultrasound (IU) and direct ultrasound (DU). In the IU method, the ultraturrax homogenized samples underwent ultrasonication in an ultrasonic bath (ULTRASONIC CLEANER, Model 30, Shenzhen Jietai Ultrasonic Cleaning Equipment Co. Ltd., Shenzhen, China) (6 L capacity, ultrasonic power: 180 W; frequency: 40 kHz) for 10 min at room temperature. The samples (E-IU and W-IU) were subsequently filtered and stored at −20 °C until further analysis. In the DU method, after homogenization, the samples were subjected to extraction using an ultrasonic processor of 400 Watts of nominal output power and 60 kHz frequency (UP400St, Hielscher Ultrasonics, Teltow, Germany) coupled with an ultrasonic probe (S24d14D sonotrode, Hielscher, Teltow, Germany) at a maximum power output of 400 W and a frequency of 60 kHz for 10 min. The temperature was not controlled during this process. Following extraction, the samples (E-DU and W-DU) were filtered and stored at −20 °C until further analysis.

Two additional techniques using hot water were tested for aqueous extraction. In both cases, ultraturrax homogenization was carried out with water at 100 °C. In the first case (infusion), the homogenized sample was allowed to stand at room temperature for 10 min, then filtered and stored at −20 °C until analysis (sample W-INF). In the second case (decoction), the homogenized sample was boiled for 10 min, followed by filtration through filter paper (Whatman No. 1) and storage at −20 °C until analysis (sample W-DEC). All extractions were performed on leaves from three different plant specimens (biological replicates), and the data shown reflect the mean ± standard deviation of these three independent biological samples.

### 2.3. Analysis of Phenolic Compounds by HPLC-DAD-MS

A Waters Alliance HPLC system (Manchester, UK) equipped with a Mediterranea Sea18 reverse-phase analytical column (25 cm length × 4.6 mm i.d., 5 μm particle size; Teknokroma, Barcelona, Spain) was used. The column outlet was directly connected to a diode array detector (DAD) (Waters 996, Millipore, Manchester, UK), followed by a Waters QDa mass detector. The mass detector conditions were as follows: capillary voltage of 0.8 kV, cone voltage of 15 V, and a probe temperature of 600 °C. The gradient profile for the separation of phenolic compounds was established using solvent A (water with 1% formic acid) and solvent B (acetonitrile) as follows: the proportion of solvent B was increased from 0% to 20% over the first 20 min, then to 21% over the next 8 min, maintained at 21% for 2 min, followed by an increase to 30% over the next 10 min, then to 100% over the next 5 min, maintained at 100% for 5 min, and finally returned to the initial conditions over the next 5 min. The flow rate was 1 mL/min, and the column temperature was 30 °C. Absorption spectra for all peaks were recorded in the 200–600 nm range, and for quantification, chromatograms were acquired at 280 nm for phenolic acids, 360 nm for flavonoids, and 420 nm for naphthoquinones. The phenolic compounds were quantified using external calibration curves and reference standards. For phenolic compounds for which no commercial standard was available, quantification was performed using the calibration curve of a compound from the same phenolic group. The results were expressed in mg per 100 g of fresh walnut leaves.

Individual phenolic compounds were identified using their retention times, absorption, and mass spectrometric data. Electrospray ionization (ESI) mass spectra were obtained at ionization energies of 20 and 50 eV (negative mode) and 20 and 50 eV (positive mode), with MS scans from *m*/*z* 50 to 1000.

### 2.4. Evaluation of Walnut Leaves Bioactivities

#### 2.4.1. Experimental Conditions for *Caenorhabditis elegans* Cultivation and Maintenance

*C. elegans* strains N2 (Bristol), CB1370 [*daf-2(e1370ts*) III], GMC101 [*dvIs100(Punc-54: amyloid-ß1-42:3′UTR unc-54 + Pmtl-2::GFP)II*], and NL5901 [*pkIs2386(unc-54p::alpha-synuclein::YFP + unc-119(+))*] were obtained from the Caenorhabditis Genetics Center (University of Minnesota, Minneapolis, MN, USA). Worms were maintained on nematode growth medium (NGM) agar plates composed of 2.5% (*w*/*v*) Agar N°1 (Oxoid Limited, Basingstoke, UK), 0.25% (*w*/*v*) Pepton N-Z-Soy(R) BL4 (Sigma Aldrich, St.-Louis, MO, USA), 0.3% (*w*/*v*) NaCl, 0.0005% (*v*/*v*) cholesterol, 1 mM CaCl_2_, 1 mM MgSO_4_, and 25 mM K_2_HPO_4_/KH_2_PO_4_ buffer (pH 6.0). Plates were seeded with *Escherichia coli* OP50 as a food source, and worms were cultured at 16 °C. M9 buffer was prepared by dissolving 3 g KH_2_PO_4_, 6 g Na_2_HPO_4_, 5 g NaCl, and 1 mL of 1 M MgSO_4_ in deionized water to a final volume of 1 L, followed by autoclave sterilization.

#### 2.4.2. Preparation of Walnut Leaf Solutions

For the biological activity study, we used direct ethanol extraction. The resulting extract was concentrated at 35 °C under reduced pressure using a rotary evaporator (Büchi R-210, Flawil, Switzerland) and subsequently dissolved in 70% acetone to yield two stock solutions at concentrations of 167 mg/mL (EC1) and 1670 mg/mL (EC2). Prior to bioassays, extracts were sterilized by filtration through 0.22 μm cellulose acetate membrane filters and maintained at −20 °C. A 70% acetone solution was used as the negative control in place of walnut leaf extracts.

#### 2.4.3. Purification of Free Phenolics from Walnut Leaves

For isolation of free phenolic compounds, 100 mL of EC1 extract was evaporated to dryness at 35 °C under reduced pressure using a rotary evaporator (Büchi R-210, Flawil, Switzerland), dissolved in 30 mL of distilled water, and subjected to liquid-liquid extraction with ethyl acetate (1:1, *v*/*v*) in triplicate. The combined ethyl acetate fractions were concentrated to dryness using a Savant Speed Vac (Bio101, Vista, CA, USA) and reconstituted in 100 mL of 70% acetone. Prior to bioassays, the extract was sterilized by filtration through a 0.22 μm cellulose acetate membrane and maintained at −20 °C. A 70% acetone solution was used as the negative control in place of the free phenolics extract.

#### 2.4.4. In Vivo Toxicity

Age-synchronized L1 larvae of *C. elegans* wild-type strain N2 were obtained by bleaching gravid hermaphrodites, followed by incubation in M9 buffer for 24 h with continuous agitation at 20 °C (reference, Basic *Caenorhabditis elegans* Methods: Synchronization and Observation). Approximately 450 L1 larvae were transferred to each NGM plate (10 NGM plates) and cultured at 20 °C until reaching young adult stage (approximately 52 h post-seeding). Young adults were subsequently collected using Milli-Q water and subjected to three consecutive washes. The nematodes were then exposed to either Milli-Q water (negative control) or varying concentrations of walnut leaf extracts in Milli-Q water in 96-well plates (48 wells per condition, 100 μL per well) for 24 h at 20 °C. The experiment was performed in triplicate. The plates were gently tapped to assess viability, and worms exhibiting movement were counted as alive. In case of doubt, the worms were touched with a picking needle. Each well contained between 10 to 20 adult worms. The results are expressed as a percentage of alive worms concerning the untreated (control) worms.

#### 2.4.5. Thermal Stress Resistance Assay

Thermotolerance was assessed by subjecting *C. elegans* to controlled thermal stress. Wild-type N2 synchronized L1 larvae (n = 200 per plate) were cultivated on NGM agar plates containing *E. coli* OP50 and supplemented with either 100 μL of EC1 (1.7 mg/mL), EC2 (17 mg/mL), free phenolics, or 70% acetone (control) until reaching late L4 stage at 20 °C. Subsequently, 30 L4 larvae were transferred to fresh NGM plates with OP50, supplemented with the same treatments. The nematodes were then exposed to thermal stress at 37 °C for 2 h, followed by a 24 h recovery period under standard culture conditions at 20 °C. Nematode viability was assessed by the absence of response to mechanical stimulation, with unresponsiveness to mechanical stimuli classified as dead. The experiment was conducted independently in triplicate.

#### 2.4.6. Oxidative Stress Induced by Juglone

Age-synchronized wild-type N2 L1 larvae (450 per plate) were cultivated on NGM plates supplemented with 100 μL of either EC1 (1.7 mg/mL), EC2 (17 mg/mL), free phenolics, or 70% acetone (control) at 20 °C until reaching adulthood (approximately 48 h). The pre-treated adult nematodes were then collected, washed three times with M9 buffer, and suspended in 15 mL falcon tubes (100 worms/mL) containing their respective treatment solutions. The suspensions were subsequently exposed to 100 μM juglone for 1 h to induce oxidative stress. Following exposure, the nematodes were washed three times with M9 buffer and transferred to fresh NGM agar plates seeded with *E. coli* OP50. Survival was assessed after 24 h, with nematodes unresponsive to mechanical stimuli considered dead. The experiment was conducted independently in triplicate.

#### 2.4.7. Fertility Assay with Daf-2(e1370) Strain

The *daf-2(e1370)* mutant contains a temperature-sensitive allele in the *C. elegans* ortholog of the insulin/insulin-like growth factor-1 (IGF-1) receptor gene. This hypomorphic allele significantly reduces fertility due to impaired insulin/IGF-1 signaling pathway activity.

Synchronized L1 larvae (n = 200 per plate) were cultivated on NGM agar plates (60 mm diameter) seeded with *E. coli* OP50 and supplemented with 100 μL of either EC1 (1.7 mg/mL), EC2 (17 mg/mL), free phenolics, or 70% acetone (control). Nematodes were maintained at 16 °C until reaching the L4 stage (approximately 2.5 days) to prevent dauer formation in the temperature-sensitive *daf-2(e1370)* mutant. Subsequently, individual L4 larvae (n = 30 per treatment group) were transferred to fresh NGM agar plates (35 mm diameter) containing identical treatments but with 50 μL extract volume and incubated at 25 °C for 72 h. Reproductive capacity was assessed by quantifying the total hatched progeny per animal. The experiment was conducted independently in triplicate.

#### 2.4.8. Paralysis Assay in GMC101 Strain

The transgenic *C. elegans* strain GMC101, engineered to express human A*β*_1_₋_42_ in its body wall muscle cells, serves as a model for Alzheimer’s disease. Elevating the culture temperature to 25 °C dramatically increases A*β* synthesis and aggregation within the worms’ musculature, ultimately leading to progressive paralysis.

Synchronized L1 larvae of the GMC101 strain (200 worms per plate) were cultured on 60 mm NGM agar plates seeded with *E. coli* OP50 and supplemented with 100 μL of either EC1 (1.7 mg/mL), EC2 (17 mg/mL), a free phenolics, or 70% acetone (control) at 16 °C until they developed to the L4 stage. After reaching L4, worms were transferred to fresh NGM plates (60 worms per treatment group, 30 per plate) containing the same treatments and incubated at 25 °C for 16 h to induce A*β*_1_₋_42_ aggregation. This temperature shift stimulates rapid amyloid deposition in the body wall muscle cells of the GMC101 strain, leading to progressive paralysis as a phenotypic outcome.

The number of paralyzed worms was scored under the microscope. To identify the paralysis, each worm was gently touched with a platinum wire. Worms were counted as paralyzed if they moved neither spontaneously nor in response to three prods on the head, at least one time their full body length.

#### 2.4.9. Locomotion Assay in NL5901 Strain

The NL5901 strain is a transgenic *C. elegans* model of Parkinson’s disease (PD), engineered to express human α-synuclein in body wall muscle cells, which results in accelerated age-related mobility decline. Three gravid hermaphrodites were placed on NGM agar plates (60 mm diameter) seeded with *E. coli* OP50 and supplemented with 100 μL of varying concentrations of walnut leaf extract, free phenolics, or 70% acetone (control) for egg-laying. Progeny was maintained at 20 °C until reaching the L4 stage (approximately 2.5 days post-hatching). Twenty synchronized L4 larvae per condition were subsequently transferred to fresh treatment plates containing identical supplementation. Locomotor function was assessed on day 5 of adulthood by quantifying body bends over a 20 s interval following a 30 s recovery period in an M9 buffer to mitigate handling-induced behavioral alterations. A minimum of 20 nematodes was analyzed per experimental condition, with experiments performed independently in triplicate as biological replicates.

### 2.5. Statistical Analysis

All samples were analyzed in triplicate (n ≥ 3). Intergroup comparisons were performed via one-way ANOVA using Statgraphics Plus program version 2.1, with *p* < 0.05 denoting statistical significance. *C. elegans* assay data were processed using GraphPad Prism 9 (v9.0a). All quantitative experiments were repeated independently in three biological replicates, yielding consistent results. Representative micrographs in figures were chosen from three independent experimental trials showing comparable progeny production and locomotion outcomes. Paralysis assay data reflect mean values derived from three independent replicates.

## 3. Results and Discussion

To identify the optimal extraction conditions for phenolic compounds from walnut leaves, this study utilized two distinct extraction media—water and 80% aqueous ethanol—paired with various extraction techniques, including indirect ultrasound and direct ultrasound. The extraction process, conducted through infusion and decoction, exclusively employed water.

### 3.1. Effect of Extraction Methods on Different Phenolic Compounds of Walnut Leaf Extracts

#### 3.1.1. Effect of Extraction Methods on Walnut Leaf Extracts’ Total Phenolic Compounds Content

The content of total phenolic compounds in walnut leaf extracts, assessed as the cumulative sum of individually quantified phenols through HPLC-DAD, varied between 144 and 1014 mg/100 g FW, influenced by the extraction methods and the solvent employed (Figure 1).

Results indicated that ethanol extracts contained more phenolic compounds, ranging from 566 to 942 mg/100 g FW, than water extracts, ranging from 140 to 344 mg/100 g FW (Figure 1). Islam et al. [34] found that among various tested solvents, such as water, methanol, ethanol, acetone, and benzene, the methanolic extract from walnut leaves exhibited the highest total phenolic content (94 mg GAE/g) compared to the aqueous extract (28 mg GAE/g). The same result was obtained in a comparative study of ethanolic and aqueous extracts of P. aphylla, Persian walnut, and oleander [35].

For samples extracted with ethanol, indirect ultrasound did not enhance the extraction of phenols, whereas direct ultrasound reduced the amount by 40%. However, in aqueous extraction, indirect ultrasound increases the extraction efficiency by around 9% (332 mg/100 g FW for W-IU compared to 305 mg/100 g FW for W). In contrast, direct ultrasound treatment reduces the extraction yield by more than 54% (140 mg/100 g FW). Hot aqueous extraction by infusion (W-INF) showed a comparable yield to indirect ultrasound extraction (344 mg/100 g FW). In contrast, decoction extraction (W-DEC) resulted in a significantly lower yield (184 mg/100 g FW).

After conducting analysis in the ethanolic and aqueous extracts, direct ultrasound treatment’s negative influence on phenolic compounds’ total content was observed. Similarly, previous research observed that direct ultrasound treatment at a power of 100 W caused the reduction of the total content of phenolic compounds by 30% compared to untreated samples of cantaloupe melon juice [36]. Moreover, others reported that sonication decreased antioxidant DPPH and FRAP capacity and total phenolic compounds in cloudy apple juice [37]. The negative effect of high-power ultrasound on polyphenolic content and antioxidant capacity was reported previously in several studies [36,37,38,39,40].

#### 3.1.2. Effect of Extraction Methods on the Total Content of Phenolic Acids, Flavonoids, and Naphthoquinones

Figure 2 evaluates the impact of the extraction method on the total content of phenolic acids, flavonoids, and naphthoquinones.

The total content of different phenolic compounds ranged from 108 to 359 mg/100 g FW for phenolic acids, from 6 to 517 mg/100 g FW for flavonoids, and from 23 to 147 mg/100 g FW for naphthoquinones. Flavonoids accounted for 50–67% of the total phenolic content in the ethanolic extracts but less than 7% in the water extracts. In contrast, phenolic acids comprised over 74% of the total phenolic content in the water extracts and between 20–43% in the ethanolic extracts. Our results align with Amaral et al. [41], who reported that more than 70% of the total phenolic content consisted of flavonoids in methanolic extracts of walnut leaves from six different cultivars, with the remainder being phenolic acids. Additionally, studies have shown that phenolic acids are the main compounds in aqueous extracts of walnut leaves, with no flavonoids detected [2]. Naphthoquinones were the minor phenolic group in the ethanolic extracts; however, they represented the second most abundant phenolic compounds in the aqueous extracts.

Depending on the extraction method and solvent used, the highest flavonoid contents, 517 and 508 mg/100 g, were obtained using E-IU and E, respectively, with 80% ethanol. This indicates that further extraction with IU had no significant effect on the extraction of flavonoids. However, additional extraction with DU reduced the flavonoid content by 24% compared to E. Water was the least effective solvent for extracting flavonoids from walnut leaves.

The highest phenolic acid content, 359 mg/100 g FW, was obtained with E using 80% ethanol and W-IU and W-INF using purified water and hot water, which yielded 274 and 281 mg/100 g FW, respectively. In the ethanolic extracts, ultrasound treatment, whether indirect (IU) or direct (DU), reduced the total phenolic acid content by 17% and 70%, respectively, compared to E. However, indirect ultrasound treatment and infusion in the water extracts increased the phenolic acid content by 11% and 14%, respectively. In contrast, direct ultrasound treatment and decoction reduced the phenolic acid content by 59% and 40%, respectively, compared to W. Therefore, the best methods for extracting phenolic acids from walnut leaves were simple homogenization using Ultra-Turrax with 80% ethanol and infusion with water. Regarding naphthoquinones, the highest content, 147 mg/100 g FW, was obtained using indirect ultrasound treatment with 80% ethanol.

It is evident that with water, neither flavonoids nor naphthoquinones are fully extracted and may be degraded during extraction. In aqueous extracts, only phenolic acids are predominantly present. The highest phenolic acid content was obtained through indirect ultrasound treatment and infusion, suggesting that oxidation processes may occur. In contrast, ethanol extraction yielded the highest levels of phenolic acids, flavonoids, and naphthoquinones through direct extraction without or with indirect ultrasound treatment. However, direct ultrasound treatment significantly reduced the content of all analyzed compounds. Similar to the findings of this study, numerous others have confirmed that high-power ultrasound treatment using an ultrasonic probe results in a more localized intensity compared to ultrasonic bath treatment, which is characterized by lower cavitation intensity and uneven distribution [42]. Increasing amplitudes can raise ultrasound intensity in most cases, potentially causing undesirable effects such as compound degradation. Additionally, higher amplitudes can erode the ultrasonic probe, leading to agitation rather than the desired cavitation and an uneven distribution of ultrasound throughout the medium [43].

### 3.2. Characterization of Phenolic Components in Walnut Leaf Extracts

Phytochemical profiling of walnut leaf extracts was performed via liquid chromatography-mass spectrometry (LC-MS), with results summarized in Table 1. The analytical methodology facilitated the tentative identification of 32 phenolic constituents, comprising 11 phenolic acids, 17 flavonoids, and 4 naphthoquinones. Structural elucidation was conducted through comparative analysis of retention times, UV absorption patterns, and negative-mode product ion spectra ([M-H]^−^) with authentic reference standards. In cases where commercial standards were unavailable, tentative structural assignments were established by comparing experimental mass fragmentation patterns with previously published spectral data.

#### 3.2.1. Phenolic Acids

Ten compounds were identified as hydroxycinnamic acids (1, 2, 3, 4, 5, 8, 9, 12, 13 and 14). Compounds 1, 2 and 5, the isomers of caffeoylquinic acid (*cis*-3-caffeoylquinic acid, *trans*-3-caffeoylquinic acid, and 4-caffeoylquinic acid) respectively have unique structural features despite sharing the same molecular mass (*m*/*z* 353) (Appendix A). Their MS^2^ fragmentation patterns are key for differentiation: *trans*-3-caffeoylquinic acid has a prominent base peak at *m*/*z* 191 (representing the quinic acid moiety) and a less intense peak at *m*/*z* 179 (from the caffeic acid moiety) (Appendix A). In contrast, *cis*-3-caffeoylquinic acid exhibits a similar pattern but with different intensity ratios (Appendix A). 4-caffeoylquinic acid is identified by a specific fragment at *m*/*z* 173, which results from the loss of water from the quinic acid fragment, appearing alongside the *m*/*z* 191 and 179 fragments with notable intensity (Appendix A). Additionally, UV spectral profiles provide further validation, with *cis*-3-caffeoylquinic acid typically showing a λₘₐₓ around 319 nm, *trans*-3-caffeoylquinic acid at 324 nm, and 4-caffeoylquinic acid revealing slight variations in absorbance intensity patterns compared to *trans*-3-caffeoylquinic acid (λₘₐₓ at 327 nm) (Table 1). These diagnostic fragmentation patterns and spectral characteristics align with those previously reported by Gutiérrez Ortiz et al. [2] and by Clifford et al. [44]. These isomers of caffeoylquinic acid have been identified previously in walnut leaves [2,13,18,45].

The four compounds 3, 4, 8, and 9 were assigned as isomers of *p*-coumaroylquinic acid based on their UV spectra (λₘₐₓ at 306–311 nm) and identical mass spectra ([M-H]^−^ at *m*/*z* 337) (Appendix A). These isomers can be differentiated by their chromatographic behavior, fragmentation patterns, and UV spectral characteristics [2]. The *trans* isomers exhibit more pronounced UV absorption compared to their *cis* counterparts, with characteristic λₘₐₓ values of 310–312 nm for *trans* isomers versus 306–307 nm for *cis* isomers (Table 1). Their MS^2^ fragmentation profiles provide definitive structural information: 3-*p*-coumaroylquinic acid isomers generate a predominant base peak at *m*/*z* 163 (*p*-coumaric acid) with a significant fragment at *m*/*z* 191 (quinic acid) (Appendix A), whereas 4-*p*-coumaroylquinic acid isomers produce a characteristic fragment at *m*/*z* 173 (resulting from dehydration of quinic acid) (Appendix A). The relative intensity ratios of these diagnostic fragments vary between geometric isomers, with *trans* forms typically exhibiting more intense *m*/*z* 163 signals compared to their *cis* analogs at equivalent elution positions. These consistent spectral and fragmentation patterns serve as reliable criteria for accurate structural elucidation and isomer discrimination. *p*-coumaroylquinic acid and their isomers have been previously identified in walnut leaves [2,45].

Compound 12 was identified as a caffeic acid derivative. This compound had a characteristic UV spectrum with λₘₐₓ at 330 nm and a typical fragmentation pattern with a daughter ion at *m*/*z* 179. This compound remains not fully characterized; however, it was already observed in the leaves [5] and in the pollen and male flowers of walnut [46,47].

Compounds 13 and 14 had similar UV spectra (λₘₐₓ at 312 nm) and exhibited a [M-H]^−^ ion at *m*/*z* 485, and its fragmentation showed fragment ions at *m*/*z* 325 [M-160]^−^ and *m*/*z* 163 [M-160-162]^−^. This compound corresponds to a *p*-coumaric acid hexoside derivative that was previously identified in walnut leave [6].

Only one hydroxybenzoic acid was identified in walnut leaves, compound 7, with the [M-H]^−^ ion at *m*/*z* 253, which suffered a neutral loss of 132 Da, yielding *m*/*z* 121, corresponding to benzoic acid. This fragmentation pattern is consistent with benzoic acid pentoside.

#### 3.2.2. Flavonoids

17 flavonoids were identified in walnut leaves, 2 flavanonols, and 15 flavonols. Compounds 18 and 21 were identified as flavanonols. Both gave [M-H]^−^ at *m*/*z* 435 with identical daughter ions at *m*/*z* 303. This fragmentation pattern is characteristic for taxifolin pentoside isomers and was observed in walnut leaves [4,5]. The other phenolic compounds corresponded to flavonols derived from quercetin (λₘₐₓ around 250 nm, MS2 fragment at *m*/*z* 301), kaempferol (λₘₐₓ around 340 nm, MS^2^ fragment at *m*/*z* 285), and myricetin (λₘₐₓ around 350 nm, MS^2^ fragment at *m*/*z* 317) (Table 1). Compounds 16, 17, 18, 19, and 20 all gave the deprotonated aglycone fragment at *m*/*z* 317, suggesting that they originated from myricetin. Compounds 16 and 17 were recognized as isomers of myricetin hexoside. These two compounds showed the loss of hexosyl moiety 162 u. Compounds 18 and 19 were identified as isomers of myricetin pentoside, which showed the loss of pentosyl moiety 132 u. Compound 20 at *m*/*z* 463, including the 317 fragments (M-H-146, loss of deoxyhexoside unit), was identified as myricetin deoxyhexoside as reported by Zhao et al. [4,48].

Compounds 21, 22, 23, 24, 25, 26, 29, and 30 all gave the deprotonated aglycone fragment at *m*/*z* 301, suggesting that they originated from quercetin. Compounds 21 and 22 at *m*/*z* 463 produced a fragment ion at *m*/*z* 301 (M-H-162, loss of hexoside moiety), was identified as isomers of quercetin-O-hexoside. Compounds 23, 24 and 25 were detected at *m*/*z* 433, fragmenting at *m*/*z* 301 (M-H-132, loss of pentosyl group). They were tentatively identified as isomers of quercetin pentoside. Compound 28 at *m*/*z* 477, including the 301 fragments (M-H-146, loss of a deoxyhexose unit), was identified as quercetin rhamnoside [4,5,6,45].

Compounds 29 and 30 showed the same negative molecular ion (*m*/*z* 489), which gave product ions at *m*/*z* 447 (loss of acetyl group) and 301 (loss of deoxyhexoside unit). Those compounds corresponding to quercetin-3-O-acetyl-deoxyhexoside isomers [5].

Compound 27 showed a pseudomolecular ion at *m*/*z* 417 and was identified as Kaempherol-pentoside due to the presence of the signals at *m*/*z* 285 (Kaempherol aglycone originating from the loss of the pentosyl moiety). Compound 28 presented a pseudomolecular ion [M-H]^−^ at *m*/*z* 431, releasing a fragment ion at *m*/*z* 285 (M-H-146, loss of a deoxyhexose unit), which might be coherent with Kaempherol-rhamnoside [4,5,6,48].

#### 3.2.3. Naphthoquinones

Four naphthoquinones were identified in walnut leaves: compounds 6, 15, 31, and 32. Compound 6 presented a pseudomolecular ion [M-H]^−^ at *m*/*z* 339, releasing a fragment ion at *m*/*z* 159 (M-H-H_2_O-180), which might be coherent with dihydroxytetralone hexoside; this compound was previously found in healthy leaves and leaves infected with Xanthomonas campestris pv. Juglandis [45]. Compound 15 showed the pseudo molecular ion at *m*/*z* 337, which gave product ions in the MS^2^ spectra at *m*/*z* 175 (corresponding to the specific fragmentation pattern of α-hydrojuglone (1,4,5-trihydroxy-1,4-naphthoquinone) and 157, thus indicating a successive loss of a hexosyl moiety (−162 Da) and water unit (H_2_O) (−18 Da) [45]. Juglone (5-hydroxy-1,4-naphthoquinone) (compound 32) was identified using a juglone standard according to its fragmentation pattern MS *m*/*z* 173 [M-H]^−^ and MS^2^ *m*/*z* 155 [M-H-H_2_O]^−^, 145 [M-H-CO]^−^, 129 [M-H-CO_2_]^−^, and 111 [M-H-CO_2_-H_2_O]^−^ [3,6,45] and Compound 31 was identified as a juglone derivative.

### 3.3. Comparison of Individual Phenolic Compounds for Walnut Leaf Extracts

32 compounds were quantified in E, E-IU, and E-DU samples (Table 2).

In the ethanolic extracts, the most abundant phenolic compound obtained by direct extraction (E) and by indirect ultrasound treatment (E-IU) was *trans*-3-caffeoylquinic acid, which represent 25% (235.1 mg/100 g FW) and 21% (200.8 mg/100 g FW) respectively of the total phenolic compounds content. The second major phenolic compounds in E and E-IU was Q-hexoside isomer 1 which represent 11% of the total content of phenolics compounds. Q-pentoside isomer 2, Q-pentoside isomer 3, Q-rhamnoside, and K-rhamnoside were also detected in E and E-IU extracts, representing between 5% and 7% of the total content of phenolic compounds. Our results are consistent with other findings reporting *trans*-3-caffeoylquinic acid as the predominant phenolic acid in walnut leaves using different methods of extraction, such as heat-assisted extractions and decoction [2,13,16,48,49]. Different major phenolic compounds have been observed in walnut leaves in various studies, including quercetin-3-O-glucoside [5,18], quercetin-3-O-rhamnoside [4], quercetin-3-galactoside [16] and *trans*-3-*p*-coumaroylquinic acid [50].

The ethanolic extracts obtained by direct ultrasound treatment contained a significant amount of Q-hexoside iso 1 (73.1 mg/100 g FW), *trans*-3-caffeolquinic acid (67.4 mg/100 g FW), and K-rhamnoside (56.7 mg/100 g FW), which respectively account for 13%, 12%, and 10% of the total content of phenolic compounds.

E-IU promotes the extraction of juglone from walnut leaves, 97 mg/100 g FW (10% of the total content of phenolic compounds), compared to E and E-DU methods, 19 mg/100 g FW (3% of the total content of phenolic compounds).

In the water extracts, *trans*-3-caffeoylquinic acid was the main phenolic acid in the different extraction methods, representing more than 49% of the total content of phenolic compounds, which is in accordance with previous findings by Ortiz et al. [2].

Juglone and juglone derivatives are not detected in the water extracts. However, indirect ultrasound treatment in water better extracts dihydroxytetralone hexoside and hydrojuglone glucoside. Their amount represents 11% of the total content of phenolic compounds compared to ethanolic extracts (less than 4%).

The extraction method greatly influences the amount of phenolic acids in the extract. Using direct ultrasound, we observed a significant decrease in the levels of phenolic acids, particularly *trans*-3-caffeoylquinic acid, by 71% compared to the direct extraction without ultrasounds. This decrease can be attributed to utilizing maximum power (400 W) for ultrasonic probes in our study. A significant decrease in caffeic acid and sinapic acid was also observed with high ultrasound power [39]. As well as the study of wang et al. [40] indicate that ultrasound accelerated the degradation and isomerization of chlorogenic acid.

Our findings demonstrate that solvent polarity plays a crucial role in determining the extraction efficiency of phenolic compounds from walnut leaves. Notably, water—a highly polar solvent—exhibited strong efficacy in extracting polar phenolic acids, particularly *trans*-3-caffeoylquinic acid, but showed limited capacity to recover less polar constituents such as flavonoids and naphthoquinones. This polarity-dependent selectivity underscores the advantage of 80% ethanol, a solvent of intermediate polarity, which produced a more comprehensive and quantitatively richer phenolic profile. The ethanol-water binary solvent system achieved an optimal polarity balance, enabling the simultaneous solubilization of both highly polar phenolic acids and moderately nonpolar flavonoids. The remarkable difference in extraction profiles between the two solvents highlights the importance of strategic solvent selection based on polarity when developing extraction protocols for bioactive compounds. The superior performance of 80% ethanol for extracting bioactive compounds from walnut leaves confirms that solvent polarity is a critical parameter that must be optimized when targeting a broad spectrum of phytochemicals with different structural characteristics and polarities.

### 3.4. In Vivo Toxicity of Walnut Leaves

The ethanolic extract obtained by direct extraction, which provided optimal conditions for phenolic compounds with the highest content and greatest diversity, was chosen to study the bioactivities of walnut leaves of the Gran Jefe variety using *C. elegans* as an in vivo model. To define the concentrations of walnut leaf ethanolic extracts to be used in the following experiments, we first assessed the toxicity of different concentrations of these plant extracts in wild-type (WT) worms *C. elegans*, a commonly used model organism in natural product bioactivity research. The worms were fed different walnut leaf concentrations, 1.7, 17, and 170 mg/mL, in aqueous dispersion for 24 h at 20 °C (Figure 3).

Lethality serves as a fundamental endpoint in determining the toxicity levels affecting nematodes. As shown in Figure 3, nematodes exposed to different walnut leaf extract concentrations (1.7, 17, and 170 mg/mL) did not show reduced viability, even at the highest tested dose. The treated *C. elegans* maintained a survival rate of 97.7%, while the viability of the control group was 98%. A significant effect was, however, detected at a much higher concentration (170 mg/mL). We found that 1.7 and 17 mg/mL of walnut leaf extracts were not toxic to nematodes, so we chose these two concentrations for the subsequent experiments.

### 3.5. Walnut Leaf Extract Enhanced Thermal and Oxidative Stress Resistance in C. elegans

The thermotolerance- and oxidative stress-resistance-enhancing potential of walnut leaf extracts was evaluated in wild-type *C. elegans* N2 worms. Thermal stress (37 °C for 2 h) and juglone-induced oxidative stress (100 µM) were applied during the L4 stage and young adulthood (2nd-day adulthood), respectively to worms exposed to walnut leaf extracts at 1.7 and 17 mg/mL during growth. Results were compared to untreated controls under identical assay conditions (Figure 4A,B).

The percentages of worm survival following the heat stress condition are presented in Figure 4.

After thermal stress, the nematodes treated with 1.7, and 17 mg/mL of the extract significantly increased their resistance to thermal stress compared to the control group. The leaf extract at a concentration of 1.7 mg/mL increases the survival of *C. elegans* by 47% and by 56% at a concentration of 17 mg/mL (Figure 4A). The same was observed after the induction of oxidative stress by juglone; treatment of worms with both concentrations significantly increased survival rate by 45% for treatment with 1.7 mg/mL and 61% after treatment with 17 mg/mL (Figure 4B). These results suggest that walnut leaf extract treatment could effectively improve the resistance of nematodes to thermal and oxidative stress.

The same conditions were used to test the effect of free phenolics isolated from walnut leaves regarding thermal and oxidative stress on *C. elegans*. The results showed that free phenolics obtained from walnut leaves extracted at a concentration of 1.7 mg/mL were proven to enhance significantly (*p* < 0.0001) the resistance of *C. elegans* to heat stress and with less extent after induction of oxidative stress. These results indicate that the effect of walnut leaf extract in improving the resistance of *C. elegans* under heat stress could be due to free phenolics. However, it did not solely contribute to the enhancement of oxidative stress. Our study demonstrated that walnut leaf extract presented beneficial effects against thermal and oxidative stress. This is the first time that the in vivo antioxidant effects of this native walnut by-product (Gran-Jefe) were reported, and we found that this effect was also true for free phenolics from walnut leaf extract. Numerous bioactive compounds found in walnut leaves are well-documented in the scientific literature for their antioxidant properties and role in regulating antioxidant defense pathways. In fact, the work of Almeida et al. [17] demonstrated that the ethanolic extracts of walnut leaves were shown to be very effective against pro-oxidant reactive species such as hydroxyl radical, superoxide radical, peroxyl radical hydrogen peroxide, and reactive nitrogen species (RNS) and peroxy-nitrite anion. Several polyphenolic constituents identified in *J. regia* leaves extracts have been previously documented to exhibit significant reactive oxygen species scavenging capacity [17]. Furthermore, various research studies have indicated that the treatment of *C. elegans* with 200 µM of quercetin or its metabolites, specifically the 3′- and 4′-O-methylated forms, significantly enhanced the resistance against thermal- and juglone-induced oxidative stress [51,52]. The same results were found following the treatment of *C. elegans* with 5-caffeoylquinic acid [53,54,55]. Many reports have highlighted the stress resistance enhancement properties of active components found in various natural compounds. Most of these active components fall into the categories of antioxidants and free radical scavengers [51,53,54,56,57].

### 3.6. The Effect of Walnut Extracts on Promoting Fertility in a Mutant of Insulin Pathway Daf-2(e1370)

The *daf-2(e1370)* mutant is a strain carrying a mutation in the DAF-2 gene, which encodes the Insulin/Insulin-like Growth Factor (IGF-1) receptor. This mutant exhibits severely reduced or absent fertility, a phenotype associated with disruptions in the insulin signaling pathway. This pathway is evolutionarily conserved among all metazoans, and studies in diverse organisms, including humans, have revealed a connection between reduced fertility and impaired insulin signaling activity—a pattern consistent with findings in *C. elegans* [58,59]. This phenotype of the reduction of fertility in *C. elegans* was confirmed by Tissenbaum and Ruvkun [59], who demonstrated that *daf-2(e1370)* mutants show markedly reduced fertility and partially penetrant embryonic lethal phenotypes. Similarly, they showed that *age-1 (mg305)* mutants, a strain with a mutation in the AGE-1 gene encoding for the catalytic subunit of the PI 3-kinase, cause the same spectrum of phenotypes.

To evaluate the impact of walnut leaf extracts on the fertility of the *daf-2(e1370)* mutant, progeny counts were measured in worms exposed to walnut leaf extract, free phenolics, or an untreated control group. Two concentrations of walnut leaves, 1.7 and 17 mg/mL were tested, and the results were illustrated in Figure 5A,B.

Walnut leaf extracts at concentrations of 1.7 and 17 mg/mL significantly promote the fertility of the *daf-2(e1370)* mutant. Both concentrations increase the number of progenies of *daf-2(e1370)* by 1,5 times compared to the control, with the same tendency. Increasing the concentration of walnut leaf extract didn’t improve its effect on the progeny of *daf-2(e1370)* mutants. For this reason, the effect of free phenolics extracted from walnut leaf at a concentration 1.7 mg/mL on the fertility of *daf-2(1370)* was examined, and the result showed that the progeny number also increased significantly after the treatment of *daf-2(e1370)* mutant with free phenolics (Figure 5C). Therefore, free phenolics by themselves also promote fertility in *daf-2 (e1370)*. Free phenolics from walnut leaves ethanolic extract may have potential for further investigation as candidates for managing diabetes and fertility problems associated with diabetes. Polycystic ovary syndrome (PCOS) is one of the most common causes of women’s infertility at reproductive age [60]. Although the etiology of this disease is complex with genetic and environmental factors involved, this syndrome is often associated with metabolic disease like type II diabetes mellitus which 80% of women with PCOS later life safer type II diabetes and insulin resistance that is generally due to the lack or reduction of insulin signaling caused by mutations or posttranslational modifications in the insulin receptor or downstream elements of the insulin-signaling cascade [58,60,61]. It’s important to mention that when comparing the genomes of humans and *C. elegans*, it’s evident that many human disease pathways are highly conserved in *C. elegans* [62]. One of those pathways is the insulin signaling pathway. In the nematode *C. elegans*, mutation of the *daf-2* gene has reduced fertility, which could be considered a model for PCOS. Currently, there is scarce information regarding the possible use of walnut leaf extract as an alternative treatment for fertility problems associated with diabetes. The previous work of Kalgaonkar et al. [63] showed that consuming walnut kernels improves metabolic and endocrine parameters in PCOS. They decreased low-density lipoprotein cholesterol, apolipoprotein B, and glycosylated hemoglobin (HgbA1c). Additionally, they increased insulin response during oral glucose tolerance tests, adiponectin, and sex hormone-binding globulin. According to Mirzababaei et al. [26], Abdoli et al. [28], and Hosseini et al. [29], walnut leaves have been traditionally used as a cure for diabetic symptoms. Evidence from scientific research, including animal studies [27,64,65,66] and human clinical trials [11,26,28,29,49] has proved that walnut leaf extracts possess properties that effectively combat and prevent diabetes. As we mentioned, walnut leaves are rich in phenolic acids, including *trans*-3-caffeoylquinic acid, which represents 23% of the total content of phenolic compounds. This compound is a powerful antioxidant polyphenol molecule, playing a preventive and therapeutic role in various diseases caused by oxidative stress and inflammation, including diabetes. It ameliorates endothelial dysfunction in diabetic mice by activating the Nrf2 anti-oxidative pathway [67]. Recent research has revealed that 5-caffeoylquinic acid can not only improve clinical symptoms in Polycystic Ovarian Syndrome (PCOS) patients [68] to metformin activity [69]. Moreover, in our study, quercetin glycoside (Q-hexoside, Q-pentoxide, and Q-rhamnoside) represents 31% of the total content of phenolic compounds. Quercetin and its glycosides can prevent the development of diabetes mellitus by reducing oxidative stress [70,71,72].

### 3.7. Effect of Walnut Leaf Extracts to Delay the *β*-amyloid (A*β*)-Induced Paralysis of Transgenic C. elegans Model of Alzheimer’s Disease 

To investigate the neuroprotective effects of walnut leaves, we utilized a transgenic *C. elegans* model of Alzheimer’s disease (strain GMC101), which expresses human amyloid-beta (A*β*)_1–42_ peptides in body wall muscles. Maintaining the culture temperature at 25 °C enhanced A*β* accumulation in the nematodes’ muscles, inducing paralysis. To assess whether walnut leaf extracts could delay A*β*-induced paralysis, GMC101 worms were exposed to two concentrations (1.7 and 17 mg/mL) of the extract. The percentage of nematodes that remained mobile after 16 h of treatment was then measured to evaluate the protective efficacy.

The result showed that both concentrations of walnut leaf extract significantly delayed paralysis in GMC101 nematodes (*p* < 0.001) relative to the untreated control, as demonstrated in Figure 6.

The percentage of worms not paralyzed was 90% and 60% in the group treated with 17 and 1.7 mg/mL of walnut leaf extracts, respectively, compared to the control group, in which the percentage of not paralyzed worms was 13%. Furthermore, walnut leaf extracts were reported to delay the *β*-amyloid (A*β*)-induced paralysis of transgenic *C. elegans* strain GMC101 in a concentration-dependent manner. Free phenolics extracted from walnut leaves at 1.7 mg/mL concentration showed equal potency and efficacy as walnut leaf extracts at 1.7 mg/mL. Therefore, the effect of walnut leaf extracts to delay the *β*-amyloid (A*β*)-induced paralysis of transgenic *C. elegans* strain GMC101 could be due to free phenolics. The main constituent of amyloid plaques found in the brains of Alzheimer’s disease patients is fibrillar amyloid beta-protein (A*β*). Chauhan et al. [30,31] showed that walnut kernel extract may reduce the risk or delay the onset of Alzheimer’s disease. They found that it could inhibit Aß fibril formation by 90% in a concentration and time-dependent manner and defibrillate Aß preformed fibrils. No studies have been performed on the neuroprotective effect of walnut leaf extracts in an in vivo model of Alzheimer’s disease. The only in vitro assay demonstrated that walnut leaf extracts had either no or relatively low inhibition of acetylcholinesterase (AChE) and butyrylcholinesterase (BChE), the enzymes vital for Alzheimer’s disease. The inhibition of those enzymes has been shown to cause an improvement in AD symptoms [15]. Numerous in vitro and in vivo studies have elucidated the antioxidant and neuroprotective potentials of flavonoids and their mechanisms of action on Alzheimer’s Disease [73,74,75]. The role of flavonoids in Alzheimer’s disease (AD) is highlighted by their ability to inhibit key enzymes such as acetylcholinesterase and butyrylcholinesterase, as well as their involvement in preventing the aggregation of Tau protein and inhibiting *β*-secretase activity. They also target oxidative stress, inflammation, and apoptosis through the modulation of signaling pathways known to play a crucial role in cognitive and neuroprotective functions. These include but are not limited to ERK, PI3-kinase/Akt, NFKB, MAPKs, and endogenous antioxidant enzymatic systems. Embrace the diverse mechanisms of action displayed by flavonoids as they work towards preventing and mitigating the effects of Alzheimer’s Disease [74]. Our research discovered that the free phenolics from walnut leaves possess significant potential in combating oxidative stress triggered by heat shock and the oxidant juglone. Consequently, the neuroprotective properties of walnut leaves can be attributed to the flavonoids contained within the extract, which can enhance the endogenous antioxidant defenses and regulate the cellular redox balance [31].

### 3.8. Motility Impairment in NL5901 Strain Is Ameliorated in Response to Walnut Leaf Extract

The neuroprotective potential of walnut leaf extract and its free phenolics was observed in our study using the transgenic *C. elegans* model of Alzheimer’s disease, GMC101. This was demonstrated by delaying the paralysis induced by *β*-amyloid (A*β*). To further validate this effect, the NL5901 strain, a Parkinson’s disease (PD) transgenic model created by inserting human α-synuclein gene specifically expressed in muscle cells causing age-dependent mobility impairments [76], was used to check the anti-Parkinson effect of walnut leaf extracts and their free phenolics (Figure 7).

Figure 7 shows that the worms treated with leaf extract at a concentration of 1.7 mg/mL and its free phenolics significantly increased body movement by 120 and 135 body bends per minute, respectively, as compared to the control (73 bends per minute), suggesting an improvement in motility. A key finding of this study reveals that free phenolics isolated from walnut leaf extract significantly delayed α-synuclein-mediated motility impairment in a validated *C. elegans* Parkinson’s model, identifying a protective role for these compounds in neurodegeneration. These results, together with the delay in the onset of the *C. elegans* model of Alzheimer’s disease, support a positive effect of walnut leaf’s free phenolics supplementation in neurodegenerative diseases in vivo. Recent research highlights that various plant extracts [77,78] and specific bioactive compounds, such as tyrosol [79], ferulic acid [80] and 5-caffeoylquinic acid [81] demonstrate the ability to suppress α-synuclein aggregation in the *C. elegans* NL5901 strain, a well-established Parkinson’s disease model.

It has been shown that there is a correlation between the presence of α-synuclein fibrils within Lewy bodies and the formation of aggregates, which has been linked to heightened levels of oxidative stress [82,83]. Based on various in vivo experiments and studies conducted on cultured cells, it has been observed that reactive oxygen species (ROS) can initiate the aggregation of α-synuclein protein. Consequently, this process further augments the production of ROS, thereby establishing a destructive cycle that ultimately leads to neurodegeneration [84,85,86]. In this sense, our results show that treating walnut leaf extract, specifically its free phenolics, can effectively improve resistance against oxidative stress. This suggests that the previously observed delay in the onset of motility dysfunction induced by α-synuclein may be linked to the ability of free phenolics in walnut leaves to scavenge reactive oxygen species (ROS). Consequently, these free phenolics could mitigate the cellular oxidative environment, thereby reducing the formation of toxic α-synuclein species.

## 4. Conclusions

This study examined optimization strategies for phenolic extraction from Gran Jefe walnut leaves and their bioactive potential. Extraction with 80% ethanol proved more effective than water, with direct extraction or indirect ultrasound treatment yielding the highest phenolic content (primarily *trans*-3-caffeoylquinic acid and quercetin-3-hexoside). Aqueous extraction only solubilized phenolic acids, while flavonoids and naphthoquinones required ethanol. Indirect ultrasound enhanced juglone extraction, while direct ultrasound showed none or negative effects. The optimized ethanolic extract significantly enhanced thermal and oxidative stress resistance in *C. elegans* and improved pathologies associated with Alzheimer’s and Parkinson’s diseases by counteracting *β*-amyloid and α-synuclein accumulation. Furthermore, this extract could be a promising candidate for further investigation in addressing diabetes and its associated fertility complications. Free phenolics were identified as the bioactive components. These findings suggest Gran Jefe walnut leaf extracts merit further investigation for managing oxidative stress-related conditions, including diabetes, fertility disorders, and neurodegenerative diseases.

## Figures and Tables

**Figure 1 foods-14-01048-f001:**
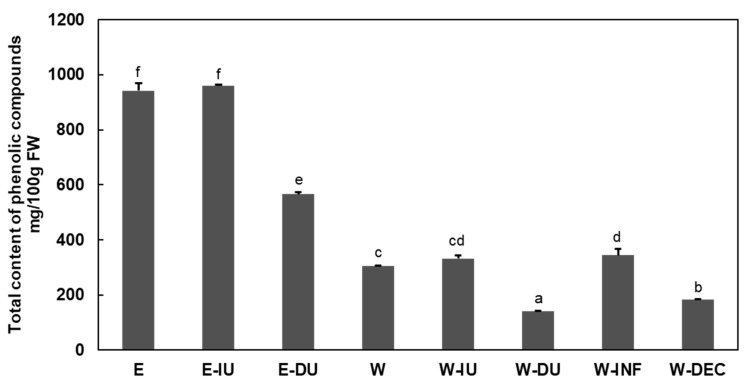
The total phenolic compounds content of obtained extracts is expressed as mg/100 g fresh weight. E: direct extraction by homogenization using ethanol 80%, E-IU: indirect ultrasound treatment using ethanol 80%, E-DU: direct ultrasound treatment using ethanol 80%, W: direct extraction by homogenization using water, W-IU: indirect ultrasound treatment using water, W-DU: direct ultrasound treatment using water, W-INF: hot aqueous extraction by infusion, W-DEC: hot aqueous extraction by decoction and FW: fresh weight. Data are presented as mean ± standard deviation (SD; n = 3). Statistically significant differences (*p* < 0.05) between experimental groups prepared using distinct methodologies are denoted by superscript letters (a–f).

**Figure 2 foods-14-01048-f002:**
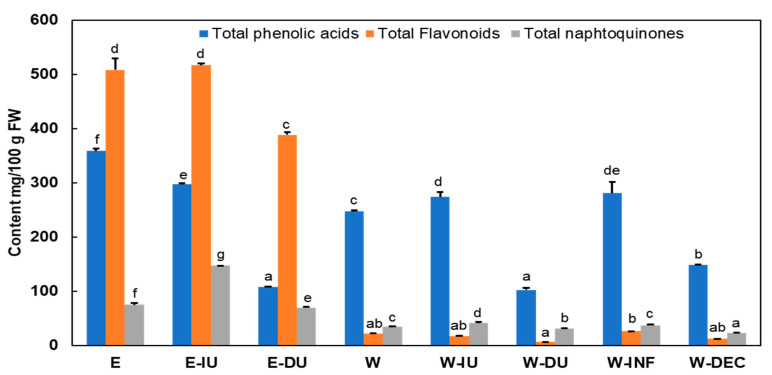
Total content of phenolic acids, flavonoids and naphthoquinones of obtained extracts expressed as mg/100 g fresh weight. E: direct extraction by homogenization using ethanol 80%, E-IU: indirect ultrasound treatment using ethanol 80%, E-DU: direct ultrasound treatment using ethanol 80%, W: direct extraction by homogenization using water, W-IU: indirect ultrasound treatment using water, W-DU: direct ultrasound treatment using water, W-INF: hot aqueous extraction by infusion, W-DEC: hot aqueous extraction by decoction and FW: fresh weight. Data are presented as mean ± standard deviation (SD; n = 3). Statistically significant differences (*p* < 0.05) between experimental groups prepared using distinct methodologies are denoted by superscript letters (a–g).

**Figure 3 foods-14-01048-f003:**
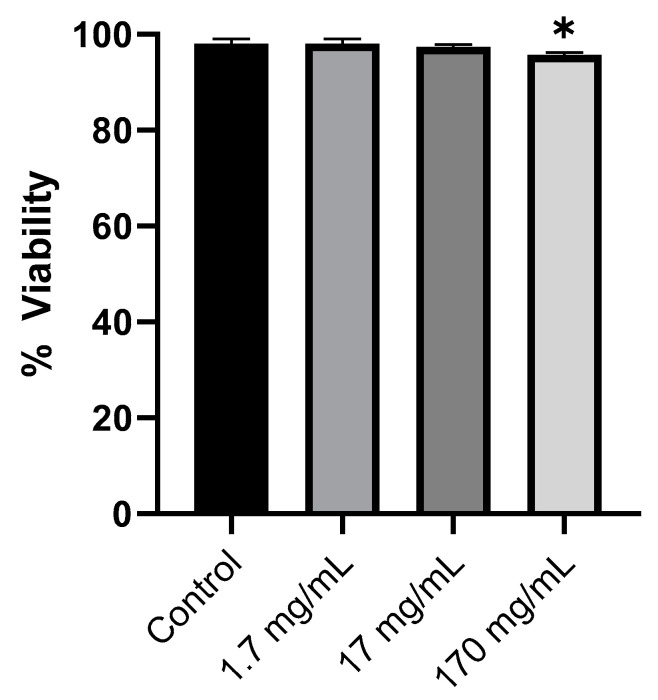
Viability of wild-type *C. elegans* (N2) following 24 h exposure to varying concentrations of walnut leaf extracts (1.7, 17, 170 mg/mL) at 20 °C. Viability data were normalized and expressed as percentages relative to untreated controls (Milli-Q water only), which were designated as 100% survival. Data are presented as mean ± standard deviation (SD) from three independent biological experiments, each comprising 48 replicates. Statistical significance was determined by one-way analysis of variance (ANOVA) followed by Dunnett’s test for multiple comparisons against the control group (* *p* < 0.05).

**Figure 4 foods-14-01048-f004:**
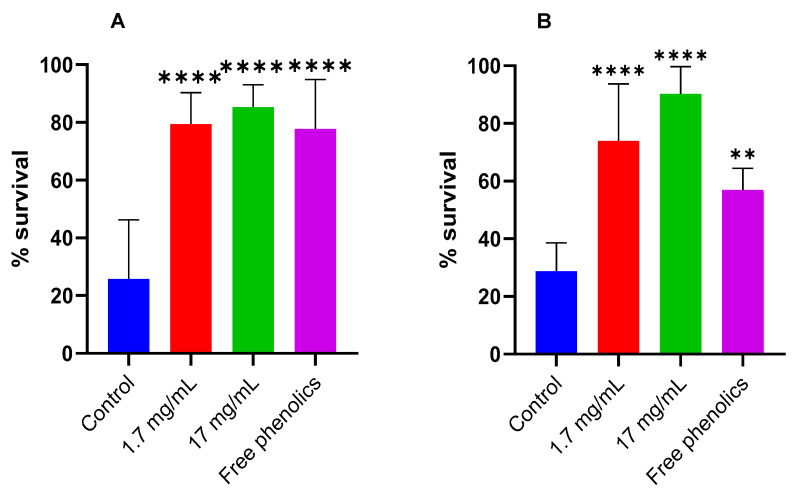
Survival rates of L4-stage *C. elegans* wild-type (N2) following exposure to (**A**) thermal stress (37 °C, 2 h) and (**B**) juglone-induced oxidative stress. Nematodes were cultivated in the absence (control) or presence of walnut leaf extract (1.7 or 17 mg/mL) or free phenolics supplementation. Data are presented as mean ± standard deviation (SD) from three independent biological replicates. Statistical significance was determined by one-way analysis of variance (ANOVA) followed by Dunnett’s test for multiple comparisons against the control group (** *p* < 0.01,**** *p* < 0.0001).

**Figure 5 foods-14-01048-f005:**
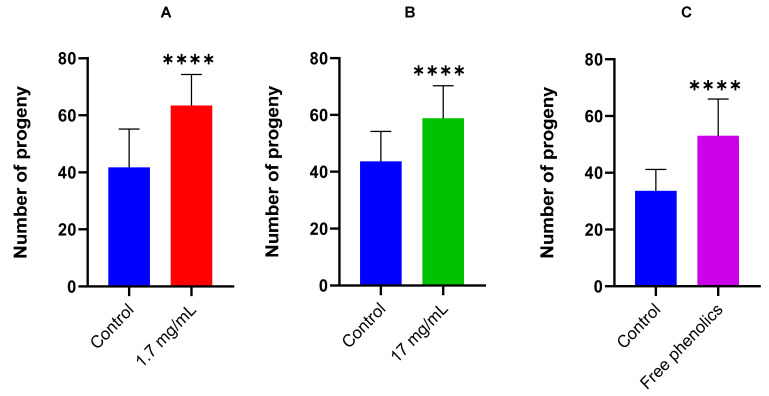
The total number of hatched offspring of the *daf-2(e1370)* strain was examined in the presence or absence (control) of walnut leaf extract at concentrations of 1.7 mg/mL (**A**) and 17 mg/mL (**B**), as well as in the presence of free phenolics extracted from the leaf extract at a concentration of 1.7 mg/mL (**C**). On the X-axis, the blue bar represents the *daf-2(e1370)* strain without any treatment, while the red bar represents the *daf-2(e1370)* strain treated with walnut leaf extract at a concentration of 1.7 mg/mL (**A**), the green bar represents the *daf-2(e1370)* strain treated with walnut leaf extract at a concentration of 17 mg/mL (**B**), and the purple bar represents the *daf-2(e1370)* strain treated with free phenolics extracted from leaves extract at a concentration of 1.7 mg/mL (**C**). The Y-axis shows the average total number of progeny produced per worm over 72 h. Treatment with walnut leaf extracts significantly increased the total number of progeny produced. Data are presented as mean ± standard deviation (n = 30 animals per condition). Statistical significance was determined by a two-tailed Student’s *t*-test (**** *p* < 0.0001). Two additional replicates with comparable results were conducted to validate these findings, and the data is provided in the Appendix A.

**Figure 6 foods-14-01048-f006:**
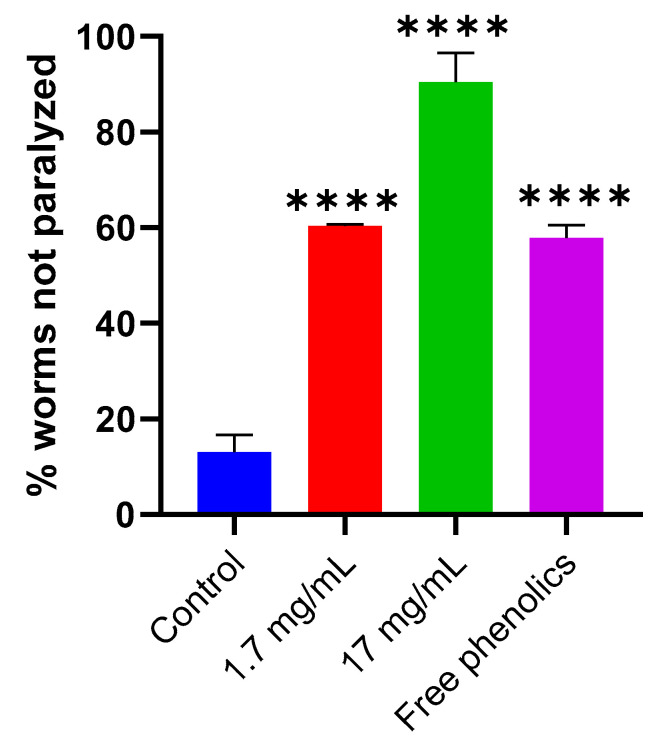
The protective effect of walnut leaf extract and free phenolics against A*β*-induced paralysis in the C. elegans GMC101 strain. This figure illustrates the impact of walnut leaf extract (1.7 mg/mL and 17 mg/mL) and free phenolics (1.7 mg/mL) on paralysis progression. The percentage of non-paralyzed worms at 25 °C is shown for untreated controls (blue bar) and treated groups: 1.7 mg/mL extract (red bar), 17 mg/mL extract (green bar), and free phenolics (purple bar). Data represent mean ± SD (n = 3 independent experiments). Statistical significance (**** *p* < 0.0001) was determined by ANOVA followed by Dunnett’s multiple comparisons test, with all groups compared to the untreated control.

**Figure 7 foods-14-01048-f007:**
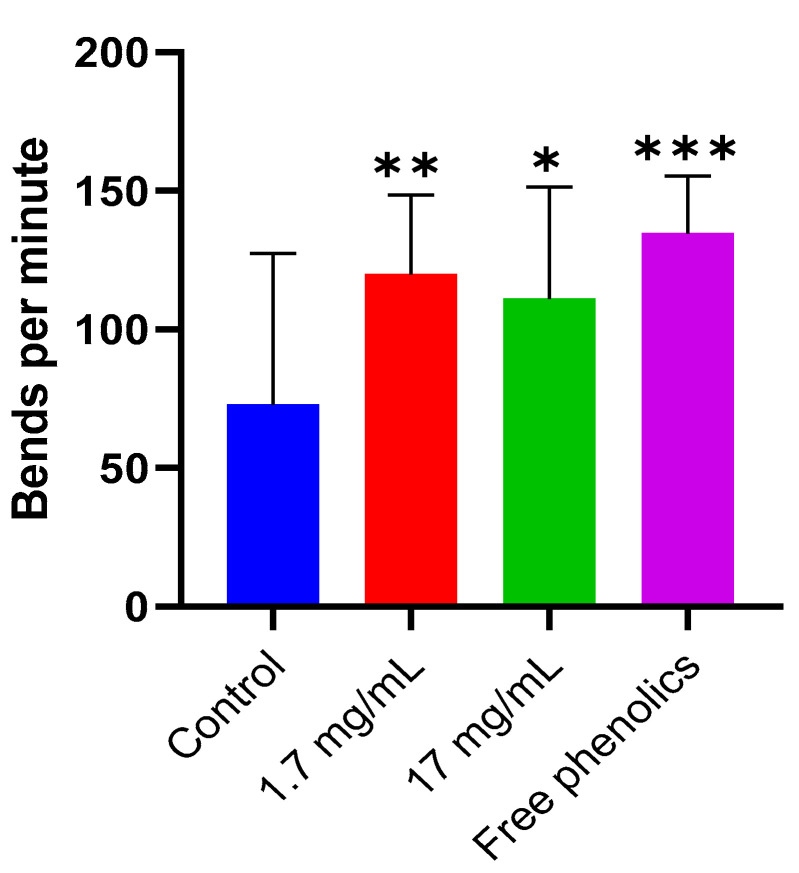
Locomotion in NL5901 worms, quantified as the number of body bends per minute, was assessed following treatment with walnut leaf extract (1.7 mg/mL or 17 mg/mL) or free phenolics (n = 20 worms per group). Graphs compare the number of bends in one minute in untreated controls (blue bar) to worms treated with 1.7 mg/mL extract (red bar), 17 mg/mL extract (green bar), and free phenolics (purple bar). Data represent mean ± SD from three independent experiments. Statistical significance (* *p* < 0.05, ** *p* < 0.01, *** *p* < 0.001) was determined by ANOVA with Dunnett’s multiple comparisons test (vs. control). One representative replicate is shown; additional replicates are provided in Appendix A.

**Table 1 foods-14-01048-t001:** Tentative identification of the 32 phenolic compounds from the leaf extracts of *J. regia* L genotype Gran Jefe.

#	Rt (min)	Compounds	λₘₐₓ	[M-H]^−^	Fragments	Equivalents
1	12.445	*cis*-3-caffeolquinic acid	319	353	191, 179, 135	Chlorogenic acid
2	13.413	*trans*-3-caffeolquinic acid	324	353	191, 179, 135	Chlorogenic acid
3	15.523	*cis*-3-*p*-coumaryolquinic acid	306	337	163, 191, 119	*p*-Coumaric acid
4	15.937	*trans*-3-*p*-coumaryolquinic acid	308	337	163, 191, 119	Chlorogenic acid
5	17.263	4-Caffeolquinic acid	327	353	173, 179, 135	Chlorogenic acid
6	18.035	Dihydroxytetralone hexoside	262, 321	339	177, 159	α-tetralone
7	18.553	Benzoic acid pentoside	307	253	121	Benzoic acid
8	20.095	*cis*-4-*p*-coumaryolquinic acid	307	337	173	*p*-Coumaric acid
9	21.068	*trans*-4-*p*-coumaryolquinic acid	311	337	173	*p*-Coumaric acid
10	29.257	Taxifolin pentoside isomer 1	290	435	303	Taxifolin
11	33.335	Taxifolin pentoside isomer 2	290	435	303	Taxifolin
12	34.177	Caffeic acid derivative	330	501	281, 179	Caffeic acid
13	37.767	Coumaric acid hexoside derivative 1	311	485	325, 163	*p*-Coumaric acid
14	39.238	Coumaric acid hexoside derivative 2	311	485	325, 163	*p*-Coumaric acid
15	19.9	Hydrojunglone glucoside	364	337	175	Juglone
16	24.602	Myrecetin hexoside isomer 1	255, 356	479	317	Myrecetin-3-O-gal
17	24.877	Myrecetin hexoside isomer 2	255, 356	479	317	Myrecetin-3-O-gal
18	24.047	Myrecetin pentoside isomer 1	255, 356	449	317	Myrecetin-3-O-gal
19	26.975	Myrecetin pentoside isomer 2	255, 357	449	317	Myrecetin-3-O-gal
20	27.338	Myrecetin rhamnoside	262, 348	463	317	Myrecetin-3-O-gal
21	28.008	Q-hexoside isomer 1	255, 350	463	301	Quercetin-3-O-glc
22	28.397	Q-hexoside isomer 2	255, 353	463	301	Quercetin-3-O-glc
23	29.837	Q-pentoside isomer 1	255, 352	433	301	Quercetin-3-O-glc
24	30.518	Q-pentoside isomer 2	255, 352	433	301	Quercetin-3-O-glc
25	31.145	Q-pentoside isomer 3	255, 350	433	301	Quercetin-3-O-glc
26	31.542	Q-rhamnoside	255, 346	447	301	Quercetin-3-O-glc
27	32.955	K-pentoside	262, 342	417	285	Nicotiflorin
28	34.675	K-rhamnoside	363, 340	431	285	Nicotiflorin
29	36.903	Q-acetyl-rhamnoside isomer 1	255, 347	489	301	Quercetin-3-O-glc
30	37.973	Q-acetyl-rhamnoside isomer 2	255, 341	489	301	Quercetin-3-O-glc
31	38.017	Juglone derivative	323, 422	489	173	Juglone
32	43.507	Juglone	349, 422	173	155	Juglone

Q: Quercetin; K: Kaempferol; gal: galactose; glc: glucose.

**Table 2 foods-14-01048-t002:** Phenolic composition of walnut leaves extracted under different conditions.

	Compound Concentration mg/100 g FW							
	Ethanol 80%			Water				
Peak Name	E	E-IU	E-DU	W	W-IU	W-DU	W-INF	W-DEC
*cis*-3-Caffeolquinic acid	15.7 ± 0.1 (2%)	16.8 ± 0.1 (2%)	7.7 ± 0.0 (1%)	3.0 ± 0.0 (1%)	18.0 ± 0.5 (5%)	6.9 ± 0.7 (5%)	18.6 ± 1.4 (5%)	8.1 ± 0.1 (4%)
*trans*-3-Caffeolquinic acid	235.1 ± 1.2 (25%)	200.8 ± 1.8 (21%)	67.4 ± 1.1 (12%)	232.2 ± 1.3 (76%)	204.6 ± 6.6 (62%)	68.4 ± 1.6 (49%)	203.6 ± 14.1 (59%)	95.5 ± 0.3 (52%)
*cis*-3-*p*-coumaryolquinic acid	8.3 ± 0.0 (1%)	4.2 ± 0.0 (0%)	1.1 ± 0.1 (0%)	0.2 ± 0.0 (0%)	4.5 ± 0.1 (1%)	1.1 ± 0.0 (1%)	4.2 ± 0.2 (1%)	2.2 ± 0.0 (1%)
*trans*-3-*p*-coumaryolquinic acid	28.4 ± 0.1 (3%)	12.4 ± 0.1 (1%)	3.0 ± 0.1 (1%)	2.2 ± 0.3 (1%)	12.2 ± 0.3 (4%)	6.4 ± 0.0 (5%)	14.7 ± 0.6 (4%)	6.5 ± 0.1 (4%)
4-caffeolquinic acid	45.3 ± 0.5 (5%)	35.4 ± 0.4 (4%)	13.5 ± 0.4 (2%)	4.5 ± 0.1 (1%)	29.7 ± 1.5 (9%)	16.4 ± 0.1 (12%)	31.2 ± 3.0 (9%)	30.6 ± 0.2 (17%)
Dihydroxytetralone hexoside	23.0 ± 1 (2%)	23.1 ± 0.1 (2%)	24.4 ± 0.9 (4%)	19.5 ± 0.0 (6%)	21.0 ± 1.2 (6%)	15.6 ± 0.3 (11%)	19.4 ± 0.8 (6%)	12.9 ± 0.6 (7%)
Benzoic acid pentoside	6.0 ± 0.6 (1%)	4.3 ± 0.1 (0%)	2.9 ± 0.1 (1%)	1.4 ± 0.0 (0%)	3.1 ± 0.2 (1%)	2.3 ± 0.2 (2%)	3.4 ± 1.4 (1%)	2.2 ± 0.1 (1%)
*cis*-4-*p*-coumaryolquinic acid	2.0 ± 0.4 (0%)	0.9 ± 0.0 (0%)	0.3 ± 0.0 (0%)	0.1 ± 0.0 (0%)	0.5 ± 0.0 (0%)	0.3 ± 0.0 (0%)	0.6 ± 0.0 (0%)	0.4 ± 0.0 (0%)
*trans*-4-*p*-coumaryolquinic acid	2.8 ± 0.1 (0%)	0.3 ± 0.0 (0%)	0.7 ± 0.0 (0%)	0.1 ± 0.0 (0%)	0.5 ± 0.0 (0%)	0.3 ± 0.0 (0%)	1.0 ± 0.0 (0%)	1.2 ± 0.0 (1%)
Taxifolin pentoside-iso 1	40.2 ± 0.1 (4%)	31.5 ± 0.5 (3%)	26.1 ± 0.2 (5%)	9.4 ± 0.1 (3%)	8.5 ± 0.3 (3%)	6.3 ± 0.0 (4%)	8.6 ± 0.0 (3%)	3.6 ± 0.1 (2%)
Taxifolin pentoside-iso 2	21.6 ± 0.1 (2%)	16.3 ± 0.5 (2%)	14.9 ± 0.2 (3%)	8.1 ± 0.0 (3%)	7.7 ± 0.1 (2%)	nd	7.5 ± 0.0 (2%)	3.2 ± 0.1 (2%)
Caffeic acid derivative	6.4 ± 0.2 (1%)	10.1 ± 0.3 (1%)	8.0 ± 0.8 (1%)	3.1 ± 0.0 (1%)	0.3 ± 0.0 (0%)	nd	2.6 ± 0.1 (1%)	1.1 ± 0.0 (1%)
Coumaric acid hexoside der 1	6.5 ± 0.5 (1%)	7.7 ± 0.1 (1%)	2.1 ± 0.1 (0%)	0.4 ± 0.0 (0%)	0.5 ± 0.0 (0%)	0.5 ± 0.0 (0%)	1.0 ± 0.0 (0%)	1.0 ± 0.0 (0%)
Coumaric acid hexoside der 2	2.4 ± 0.2 (0%)	4.4 ± 0.1 (0%)	1.6 ± 0.0 (0%)	nd	nd	nd	nd	nd
Hydrojunglone glucoside	22.2 ± 0.5 (2%)	21.4 ± 1.2 (2%)	22.1 ± 0.7 (4%)	15.4 ± 0.2 (5%)	20.1 ± 0.6 (6%)	15.5 ± 0.9 (11%)	17.9 ± 0.4 (5%)	10.4 ± 0.0 (6%)
Myrecetin hexoside iso 1	10.8 ± 0.2 (1%)	4.8 ± 0.1 (1%)	1.8 ± 0.1 (0%)	0.7 ± 0.0 (0%)	nd	nd	nd	0.3 ± 0.0 (0%)
Myrecetin hexoside iso 2	4.2 ± 0.1 (0%)	1.3 ± 0.4 (0%)	0.6 ± 0.0 (0%)	nd	nd	nd	nd	nd
Myrecetin pentoside iso 1	3.2 ± 0.1 (0%)	0.6 ± 0.0 (0%)	0.6 ± 0.0 (0%)	nd	nd	nd	nd	nd
Myrecetin pentoside iso 2	1.8 ± 0.1 (1%)	1.0 ± 0.0 (0%)	1.0 ± 0.0 (0%)	nd	nd	nd	0.7 ± 0.0 (0%)	0.3 ± 0.0 (0%)
Myrecetin rhamnoside	9.5 ± 0.2 (1%)	3.8 ± 0.3 (0%)	1.1 ± 0.1 (0%)	nd	nd	nd	nd	nd
Q-hexoside iso 1	99.4 ± 4.5 (10%)	106.6 ± 1.4 (11%)	73.1 ± 1.2 (13%)	nd	1.0 ± 0.0 (0%)	nd	6.1 ± 0.0 (2%)	3.3 ± 0.1 (2%)
Q-hexoside iso 2	32.1 ± 1.5 (3%)	26.3 ± 0.5 (3%)	17.6 ± 0.3 (3%)	nd	nd	nd	nd	nd
Q-pentoside iso 1	9.1 ± 0.6 (1%)	12.2 ± 0.2 (1%)	8.1 ± 0.2 (1%)	nd	nd	nd	nd	nd
Q-pentoside iso 2	55.7 ± 3.4 (5%)	63.2 ± 1.4 (7%)	46.3 ± 1.2 (8%)	nd	nd	nd	nd	nd
Q-pentoside iso 3	61.2 ± 3.2 (6%)	71.9 ± 0.3 (7%)	50.8 ± 1.6 (9%)	nd	nd	nd	nd	nd
Q-rhamnoside	59.3 ± 1.4 (6%)	56.3 ± 0.8 (6%)	44.5 ± 1.7 (8%)	0.2 ± 0.0 (0%)	nd	nd	1.7 ± 0.1 (0%)	0.9 ± 0.0 (0%)
K-pentoside	35.3 ± 1.3 (4%)	43.4 ± 0.9 (5%)	38.7 ± 0.1 (7%)	1.6 ± 0.1 (1%)	nd	nd	0.5 ± 0.0 (0%)	0.3 ± 0.0 (0%)
K-rhamnoside	59.5 ± 4.3 (6%)	66.8 ± 0.4 (7%)	56.7 ± 1.8 (10%)	2.4 ± 0.0 (1%)	nd	nd	0.8 ± 0.0 (0%)	0.2 ± 0.0 (0%)
Q-acetyl-rhamnoside iso 1	4.1 ± 0.3 (0%)	8.8 ± 0.8 (1%)	4.9 ± 0.1 (1%)	nd	nd	nd	nd	nd
Q-acetyl-rhamnoside iso 2	1.1 ± 0.3 (0%)	1.7 ± 0.1 (0%)	1.3 ± 0.0 (0%)	nd	nd	nd	nd	nd
Juglone der	10.0 ± 0.6 (1%)	5.3 ± 0.1 (1%)	4.1 ± 0.1 (1%)	nd	nd	nd	nd	nd
Juglone	19.6 ± 2.2 (2%)	97.2 ± 1.0 (10%)	19.0 ± 0.3 (3%)	nd	nd	nd	nd	nd

Data mg/100 g fresh leaves are the mean of three replicates. Results correspond to the mean ± standard deviation. E: direct extraction by homogenization using ethanol 80%, E-IU: indirect ultrasound treatment using ethanol 80%, E-DU: direct ultrasound treatment using ethanol 80%, W: direct extraction by homogenization using water, W-IU: indirect ultrasound treatment using water, W-DU: direct ultrasound treatment using water, W-INF: hot aqueous extraction by infusion, and W-DEC: hot aqueous extraction by decoction. FW: fresh weight, nd: not detected, Q: Quercetin; K: Kaempferol; iso: isomer; der: derivative.

## Data Availability

The original contributions presented in the study are included in the article/Appendix A, further inquiries can be directed to the corresponding authors.

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
