# Peer review of "Harnessing the Potential of Walnut Leaves from Nerpio: Unveiling Extraction Techniques and Bioactivity Through Caenorhabditis elegans Studies"

_foods, 2025, doi:10.3390/foods14061048_

Round 1
Reviewer 1 Report
Comments and Suggestions for Authors
1) Make figures 1 and 2 clearer, the letters on the histogram bars are not easy to read
2) The structural characterization of the phenolic component is not certain for the pool of compounds. Please include the following work in the references and specify better https://doi.org/10.1016/j.foodchem.2007.05.081
3)I can't find how many nematodes were used for the in vivo tests and the number of replicates, please specify
4) I think it is risky to indicate the extract as a useful treatment for diabetes and diabetes-related infertility. The results are very preliminary. Please tone down the claims
Author Response
Comment 1: Make figures 1 and 2 clearer, the letters on the histogram bars are not easy to read
Response 1: Done
Comment 2: The structural characterization of the phenolic component is not certain for the pool of compounds. Please include the following work in the references and specify better https://doi.org/10.1016/j.foodchem.2007.05.081
Response 2: We agree with the reviewer's comment. In response, we have included detailed structural characterization of the isomers of caffeoylquinic acid (page number 9; lines 425-441) and p-coumaroylquinic acid (page number 10-11; lines 446-461). This includes comprehensive information about the distinguishing features in MS² fragmentation patterns and UV spectral characteristics that allow for reliable differentiation between isomers. Additionally, we have incorporated the recommended work by Amaral et al. (2008) into our references.
Comment 3: I can't find how many nematodes were used for the in vivo tests and the number of replicates, please specify.
Response 3: Thank you for highlighting this concern. As indicated in the Materials and Methods section, we have specified the number of nematodes used for each in vivo test. For the toxicity assay, approximately 10-20 adult worms were used per well with 48 wells per condition (section 2.4.4). For thermal stress resistance, 30 L4 larvae were used per treatment group (section 2.4.5). In the oxidative stress assay, 100 worms/mL were used (section 2.4.6). For the fertility assay with daf-2(e1370) strain, 30 worms were tested per group (section 2.4.7). In the paralysis assay, 60 worms per group were used, with 30 worms per plate (section 2.4.8). For the locomotion assay, 20 animals were analyzed per experiment (section 2.4.9). All experiments were performed at least in triplicate as stated at the end of each experimental procedure section.
Comment 4: I think it is risky to indicate the extract as a useful treatment for diabetes and diabetes-related infertility. The results are very preliminary. Please tone down the claims.
Response 4: Thank you for this important feedback. We have revised the manuscript to moderate our claims. The statement on page 18 (line 717) has been changed from "could be a good treatment" to "may have potential for further investigation as candidates for managing". Similarly, the statement in the conclusion section (page 21, lines 859-860) has been changed from "it could be a promising treatment for" to "could be a promising candidate for further investigation in addressing diabetes and its associated fertility complications ".
Reviewer 2 Report
Comments and Suggestions for Authors
Observations and suggestions
-
Remove the point from the title
-
Line 23 in vivo in italics
-
Line 32 leave space in nuts[1] and correct the point
-
Line 35 leave space in tannins[7 and fiber[9]
-
Line 36 leave space in properties[5
-
Line 40 leave space in benefits[2, review the entire document as this observation is very frequent
-
Adjust the objective with the title of the research
-
Change minutes to min throughout the document, for example on line 121
-
On line 130 mention the measurements and conditions of the filter paper
-
In the chemical formulas throughout the document, place the numbers in subscript, for example CaCl2 MgSO4 (line 164)
-
Line 165 leave space K2HPO4/KH2PO4(25 and place the numbers in subscript
-
Line 167 change liter to L
-
Line 169 remove whitespace in extraction. It
-
Line 172 change ml to mL (check throughout document)
-
Line 177 change 35 ºC (check the entire document)
-
Line 184 change In vivo to italics (review entire document)
-
Line 187 remove double parenthesis
-
Line 202 correct °C
-
Line 215 leave space in 1h (check the entire document)
-
Change hours to h throughout the document, for example line 216
-
Change seconds to its abbreviation line 259
-
Check on the line if it is correct from 942 to 566 mg/100g or 566 to 942 mg/100g as well as 344 to 140 mg/100g
-
On line 289 put the word P. aphylla in italics
-
Sometimes it is reported in the document ([M-H]-) and in others ([M – H]–
-
Line 393 is missing a point
-
On line 408 correct the point
-
On line 411 check if the _ is correct
-
In line 413 remove the point
-
On line 423 capitalize t
-
On line 443 remove the double parentheses
-
On line 446 check if the CO molecule is reported correctly
-
On line 455 place the legend below table 2
-
In line 469 there is a missing point
-
On line 479 correct the point
-
On line 511 start with a capital letter and leave a space
-
On line 549 correct (P= ≤0.0001)
-
A point is missing on line 562
-
On line 566 check if the word thermal- is correct
-
On line 572 the word daf2(e1370) has no space and on the one found on line 574 it does, match in the document
-
On line 605 remove the double parentheses
-
On line 628 correct the point
-
On line 687, put the words in vitro and in vivo in italics
-
Check that all results have their corresponding discussion
-
The conclusion can be reduced
-
Check that the references comply with the journal's requirements
-
Equalize the space between units in the document

Author Response
Comment 1: Remove the point from the title
Response 1: Done
Comment 2: Line 23 in vivo in italics
Response 2: We have italicized "in vivo" in line 23 and have ensured consistent italicization of this term throughout the entire manuscript.
Comment 3: Line 32 leave space in nuts[1] and correct the point
Response 3: Done
Comment 4: Line 35 leave space in tannins[7 and fiber[9]
Response 4: Done
Comment 5: Line 36 leave space in properties[5
Response 5: Done
Comment 6: Line 40 leave space in benefits[2, review the entire document as this observation is very frequent
Response 6: Done
Comment 8: Change minutes to min throughout the document, for example on line 1
Response 8: Done
Comment 9: On line 130 mention the measurements and conditions of the filter paper
Response 9: Done.
Comment 10: In the chemical formulas throughout the document, place the numbers in subscript, for example CaCl2 MgSO4 (line 164)
Response 10: Done
Comment 11: Line 165 leave space K2HPO4/KH2PO4(25 and place the numbers in subscript
Response 11: Done
Comment 12: Line 167 change liter to L
Response 12: Done
Comment 13: Line 169 remove whitespace in extraction. It
Response 13: Done
Comment 14: Line 172 change ml to mL (check throughout document)
Response 14: Done
Comment 15: Line 177 change 35 ºC (check the entire document)
Response 15: Done
Comment 16: Line 184 change In vivo to italics (review entire document)
Response 16: Done
Comment 17: Line 187 remove double parenthesis
Response 17: Done
Comment 18: Line 202 correct °C
Response 18: Done
Comment 19: Line 215 leave space in 1h (check the entire document)
Response 19: Done
Comment 20: Change hours to h throughout the document, for example line 216
Response 20: Done
Comment 21: Change seconds to its abbreviation line 259
Response 21: Done
Comment 22: Check on the line if it is correct from 942 to 566 mg/100g or 566 to 942 mg/100g as well as 344 to 140 mg/100g
Response 22: Done
Comment 23: On line 289 put the word P. aphylla in italics
Response 23: Done
Comment 24: Sometimes it is reported in the document ([M-H]-) and in others ([M – H]–
Response 24: We have standardized the notation throughout the manuscript to consistently use ([M-H]-) for deprotonated molecular ions.
Comment 25: Line 393 is missing a point
Response 25: Done
Comment 26: On line 408 correct the point
Response 26: Done
Comment 27: On line 411 check if the _ is correct
Response 27: We have removed the underscore character “_ “from line 411.
Comment 28: In line 413 remove the point
Response 28: Done
Comment 29: On line 423 capitalize t
Response 29: Done
Comment 30: On line 443 remove the double parentheses
Response 30: Done
Comment 31: On line 446 check if the CO molecule is reported correctly
Response 31: We have verified the chemical formula and confirm that the CO molecule is correctly reported on line 446.
Comment 32: On line 455 place the legend below table 2
Response 32: Done
Comment 33: In line 469 there is a missing point
Response 33: Done
Comment 34: On line 479 correct the point
Response 34: Done
Comment 35: On line 511 start with a capital letter and leave a space
Response 35: Done
Comment 36: On line 549 correct (P= ≤0.0001)
Response 36: Done
Comment 37: A point is missing on line 562
Response 37: Done
Comment 38: On line 566 check if the word thermal- is correct
Response 38: We have verified that "thermal-" with the hyphen is correct in this context, as it forms part of the compound modifier "thermal- and juglone-induced oxidative stress."
Comment 39: On line 572 the word daf2(e1370) has no space and on the one found on line 574 it does, match in the document
Response 39: We have standardized the formatting throughout the manuscript to consistently use "daf-2(e1370)" with a hyphen between "daf" and "2" and no space between "2" and the parenthesis. This format follows the standard C. elegans gene nomenclature guidelines.
Comment 40: On line 605 remove the double parentheses
Response 40: Done
Comment 41: On line 628 correct the point
Response 41: Done
Comment 42: On line 687, put the words in vitro and in vivo in italics
Response 42: Done
Comment 43: Check that all results have their corresponding discussion
Response 43: Done
Comment 44: The conclusion can be reduced
Response 44: Done
Comment 45: Check that the references comply with the journal's requirements
Response 45: Done
Comment 46: Equalize the space between units in the document
Response 46: Done
Reviewer 3 Report
Comments and Suggestions for Authors
The authors of this paper evaluated phenolic extraction from 'Gran Jefe' walnut leaves using ethanol and water. Direct ethanol extraction was the most effective, yielding compounds like caffeoylquinic acid and quercetin-hexoside. The extract improved stress resistance, fertility, and neuroprotection in C. elegans, showing potential for Alzheimer's and Parkinson's models. I believe the paper is publishable, but it requires major revision.
Add the scientific name of the plant in the abstract.
There is a difference between “samples“ and replicates. How many samples were used for extraction (Figure 1)? The samples should come from different plant specimens.
In this type of research with plants, samples should be freeze-dried and not used fresh. Moisture content can vary significatively during sample processing.
Include the MS spectra and the corresponding fragmentation patterns as Supplementary Material.
Table 2 can be improved. Please separate “100 g”. Abbreviations should appear as footnotes.
Figure 3 has a problem. Are you sure that the concentration is 420 mg/mL? This must be over saturated. You are probable talking about 420 µg/mL.
Figure 4. Use just p<0.05. Check the units for concentration. ANOVA? Check homoscedasticity. The same for Figure 5. Number of progeny is eggs?
English should be extensively revised.
Be careful “ml” is not correct. Always use “mL”.
Figures 6 and 7. Avoid using different colors for the same chemical or extract. The control group in Figure 7 is not good. Please add a greater n.
The conclusion is long, try to reduce it.
Comments on the Quality of English LanguageIt must be improved.
Author Response
Comment 1: Add the scientific name of the plant in the abstract.
Response 1: Done
Comment 2: There is a difference between “samples“ and replicates. How many samples were used for extraction (Figure 1)? The samples should come from different plant specimens.
Response 2: In our study, leaves from three different plant specimens of the Gran Jefe variety, collected from the Nerpio plantation, were used. For all of these extraction methods (E, E-IU, E-DU, W, W-IU, W-DU, W-INF, and W-DEC), we prepared extracts from each of these three plant specimens independently. The values reported in Figure 1 represent the mean ± standard deviation of these three biological samples (n=3), each representing a different plant. We have revised the Materials and Methods section (2.2. Extraction Procedure) to clarify this important distinction, adding: " All extractions were performed on leaves from three different plant specimens (biological replicates), and the data shown reflect the mean ± standard deviation of these three independent biological samples."
Comment 3: In this type of research with plants, samples should be freeze-dried and not used fresh. Moisture content can vary significatively during sample processing.
Response 3: We appreciate your comment regarding sample preparation methodology. While freeze-drying is indeed an excellent approach for plant material preservation, we specifically chose to use fresh leaves in this study for several reasons. Our research aimed to evaluate the extraction of bioactive compounds under conditions that would be most accessible for practical applications (such as in food and nutraceutical industries) where fresh leaves are commonly used. Additionally, we were particularly interested in preserving the natural state of certain compounds (especially naphthoquinones like juglone) that can be altered during freeze-drying processes.
Comment 4: Include the MS spectra and the corresponding fragmentation patterns as Supplementary Material.
Response 4: Thank you for this constructive suggestion. We have added the MS spectra of the isomers of caffeolquinic acid and p-coumaryolquinic acid along with their corresponding fragmentation patterns as Supplementary Material. These spectra provide visual confirmation of our structural assignments and will be valuable to readers interested in the detailed chemical characterization of these compounds. The supplementary figures include both the full MS spectra and the MS² fragmentation patterns for each isomer, with peak assignments clearly labeled.
Comment 5: Table 2 can be improved. Please separate “100 g”. Abbreviations should appear as footnotes.
Response 5: Done
Comment 6: Figure 3 has a problem. Are you sure that the concentration is 420 mg/mL? This must be over saturated. You are probable talking about 420 µg/mL.
Response 6: Thank you for your important question regarding the concentration in Figure 3. After careful review of our calculations, we have identified and corrected an error in our concentration reporting. The highest concentration used should be 170 mg/mL (not 420 mg/mL as previously stated). This concentration was specifically selected to establish the upper toxicity threshold in our model organism. We have corrected this calculation error throughout the manuscript to ensure accuracy and consistency in our reporting. Thank you for prompting this important verification.
Comment 7: Figure 4. Use just p<0.05. Check the units for concentration. ANOVA? Check homoscedasticity. The same for Figure 5. Number of progeny is eggs?
Response 7: Thank you for your detailed comments regarding the statistical analysis presentation in our figures.
Regarding Figure 4: The data were processed using GraphPad Prism 9 (v9.0a). Statistical significance (*p = 0.05, **p ≤ 0.01, ***p ≤ 0.001) was determined by one-way analysis of variance (ANOVA) followed by Dunnett's test for multiple comparisons against the control group. We have double-checked and confirmed that the concentrations are correctly reported as mg/mL.
Regarding Figure 5: Statistical analysis was performed using GraphPad Prism 9 (v9.0a), with significance determined by a two-tailed Student's t-test. (****p < 0.0001). To clarify your question about "number of progeny" - this represents the total number of hatched offspring (not eggs) produced by each worm during the 72-hour test period.
Comment 8: English should be extensively revised.
Response 8: We have revised the entire text.
Comment 9: Be careful “ml” is not correct. Always use “mL”.
Response 9: Done
Comment 10: Figures 6 and 7. Avoid using different colors for the same chemical or extract. The control group in Figure 7 is not good. Please add a greater n.
Response 10: Sorry, we didn´t understand the suggestion.
Comment 11: The conclusion is long, try to reduce it.
Response 11: Done
Reviewer 4 Report
Comments and Suggestions for Authors
The research from the manuscript is solid, and the authors investigated various aspects regarding walnut leaf extracts, from extraction optimization to bioactivity. C. elegans is an excellent in vivo model, as the authors pointed out in their paper, who used it for toxicity assessment and the influence on the pathology of neurodegenerative disorders or infertility.
The Introduction clearly presents the previously published data and states the aim of the current research.
The Materials and Methods section is detailed. I have one question: why are these particular concentrations - 167 mg/ml (EC1) and 1670 mg/ml (EC2) in section 2.4.2, and 4,2 mg/ml; 42 mg/ml and 420 mg/ml for bioactivity assessment?
The results are presented in comparison to other similar studies, and the discussion section is solid.
The Conclusions are supported by the results, but this section is too long, in my opinion, and it could be rephrased. Lines 763-765 should be moved to the Results and Discussions section, explaining why only E extracts were used for bioactivity testing.
There are also some writing issues: lines 417 (reported by ... author), 429, 511.
Author Response
Comment 1: The Materials and Methods section is detailed. I have one question: why are these particular concentrations - 167 mg/ml (EC1) and 1670 mg/ml (EC2) in section 2.4.2, and 4,2 mg/ml; 42 mg/ml and 420 mg/ml for bioactivity assessment?
Response 1: Thank you for this important question regarding our concentration calculations. The concentrations 167 mg/ml (EC1) and 1670 mg/ml (EC2) represent our stock solutions. For the bioactivity assessment, we applied 100 μL of these stock solutions to plates containing 10 mL NGM agar, resulting in final working concentrations of 1,7 mg/ml and 17 mg/ml respectively. Upon careful review, we identified a calculation error in our manuscript regarding the final concentrations. We have corrected these values throughout the manuscript, changing 4.2 mg/mL to 1.7 mg/mL, 42 mg/mL to 17 mg/mL, and 420 mg/mL to 170 mg/mL. These corrections have been applied consistently across all relevant sections including methods, results, figure captions, and discussion. We appreciate you bringing this to our attention, allowing us to ensure the accuracy of our reported concentrations.
Comment 2: The Conclusions are supported by the results, but this section is too long, in my opinion, and it could be rephrased.
Response 2: We changed the conclusion paragraph to “This study examined optimization strategies for phenolic extraction from Gran Jefe walnut leaves and their bioactive potential. Extraction with 80% ethanol proved more effective than water, with direct extraction or indirect ultrasound treatment yielding the highest phenolic content (primarily trans-3-caffeoylquinic acid and quercetin-3-hexoside). Aqueous extraction only solubilized phenolic acids, while flavonoids and naphthoquinones required ethanol. Indirect ultrasound enhanced juglone extraction, while direct ultrasound showed none or negative effects. The optimized ethanolic extract significantly enhanced thermal and oxidative stress resistance in C. elegans and improved pathologies associated with Alzheimer's and Parkinson's diseases by counteracting β-amyloid and α-synuclein accumulation. Furthermore, this extract could be promising candidates for further investigation in addressing diabetes and its associated fertility complications. Free phenolics were identified as the bioactive components. These findings suggest Gran Jefe walnut leaf extracts merit further investigation for managing oxidative stress-related conditions including diabetes, fertility disorders, and neurodegenerative diseases.”
Comment 3: Lines 763-765 should be moved to the Results and Discussions section, explaining why only E extracts were used for bioactivity testing.
Response 3: Thank you for this suggestion. We have moved the content from lines 763-765 to the Results and Discussion section, specifically to the beginning of section 3.4, to provide a clear rationale for our extract selection for bioactivity testing. The added sentence reads: "The ethanolic extract obtained by direct extraction, which provided optimal conditions for phenolic compounds with the highest content and greatest diversity, was chosen to study the bioactivities of walnut leaves of the Gran Jefe variety using C. elegans as an in vivo model." This addition helps establish a logical connection between our extraction optimization findings and the subsequent bioactivity studies.
Comment 4: There are also some writing issues: lines 417 (reported by ... author), 429, 511.
Response 4: Corrected
Round 2
Reviewer 3 Report
Comments and Suggestions for Authors
The paper improved a lot.
Still several things to correct.
Figures 1 and 2. p<0,05, use italics for letter p. Error bars can be improved, better line (increase width).
Figure 3 can be improved. Asterisk in the center, separated from the error bar.
Figures 4, 5, 6 and 7. Use only p<0.05.
Use the same colors for the columns.
The corrections are mandatory.
Author Response
Comment 1: Figures 1 and 2. p<0,05, use italics for letter p. Error bars can be improved, better line (increase width).
Response 1: Done
Comment 2: Figure 3 can be improved. Asterisk in the center, separated from the error bar.
Response 2: Done
Comment 3: Figures 4, 5, 6 and 7. Use only p<0.05.
Response 3: We appreciate the reviewer's suggestion to simplify the statistical notation in Figures 4, 5, 6 and 7. After careful consideration, we respectfully propose maintaining the multiple significance levels (*, **, ***, ****) in these figures for the following reasons:
Informational value: The differentiated significance notations provide readers with more precise information about the statistical strength of our findings. This is particularly important in our study, where the magnitude of effects varies substantially between treatments and experimental conditions.
Field convention: Multiple significance levels are commonly used in publications within our research area, including recent papers in Biomedicine & Pharmacotherapy: https://www.sciencedirect.com/science/article/pii/S0753332224015634?via%3Dihub
International Journal of Molecular Sciences: https://doi.org/10.3390/ijms241512079
Foods: https://doi.org/10.3390/foods12010081
This approach aligns with standard practices in bioactivity research using C. elegans as model organisms.
Results interpretation: Several of our key findings show notably strong statistical significance (p<0.0001), which we believe is important contextual information for readers evaluating the robustness of walnut leaf extract effects. The graduated system allows readers to quickly distinguish between marginally significant results and highly significant ones.
Consistency with methods: Our statistical analysis section details the multiple comparison procedures used, which naturally produce these varied significance levels.
However, to address the reviewer's concern about clarity, we have modified the figure legends to more clearly explain the significance notation (e.g., "Statistical significance indicated as *p<0.05, **p<0.01, ***p<0.001, ****p<0.0001") and have ensured consistent application across all figures.
We believe this approach strikes a balance between providing detailed statistical information while maintaining clarity, but we remain open to further discussion if the reviewer has additional concerns.
Comment 4: Use the same colors for the columns.
Response 4: Done